# Near-unity CO$_2$-to-ethylene photoconversion over low coordination single-atom catalysts

Zhiling Tang[1,8], Yingli Wang[1,8], Tian Qin[2], Yuechang Wei [1] ✉, Jing Xiong[1], Xiong Wang[1], Xuanzhen Li[1], Min Liu [3,4] ✉, Yunpeng Liu [5,6] ✉, Xi Liu [2,7] ✉ & Zhen Zhao [1]

Photocatalytic conversion of carbon dioxide to value-added chemicals, particularly multi-carbon products, offers a promising route toward carbon-neutral cycles. However, achieving high activity and selectivity remains extremely challenging due to the instability of key reaction intermediates and limited C–C coupling efficiency. Herein, we report a low-coordination manganese single-atom catalyst embedded in zinc sulfide (Mn$_1$–ZnS$_v$) that enables efficient and selective CO$_2$-to-C$_{2+}$ conversion. In-situ spectroscopic analyses and density functional theory calculations reveal that sulfur vacancies are created at the Mn single-atom coordination sites and induce the formation of coordination-unsaturated Mn-S$_2$ configuration. The asymmetric coordination environment of Mn modulates local charge distribution, strengthens *CO adsorption, and promotes *CO and *CHO coupling to form the *COCHO intermediate for efficient C–C coupling. As a result, the Mn$_1$–ZnS$_v$ catalyst achieved 99.1% selectivity for ethylene with a formation rate of 76.6 µmol g$^{-1}$ h$^{-1}$. This study highlights the critical role of atomic-level coordination engineering in advancing photocatalytic CO$_2$-to-C$_{2+}$ conversion.

Solar-driven conversion of carbon dioxide (CO$_2$) into multi-carbon (C$_{2+}$) solar fuels has garnered significant attention as a promising green solution to address global energy demands and mitigate the impacts of climate change[1–3]. Although recent advances have demonstrated the potential of photocatalytic systems for CO$_2$-to-C$_{2+}$ conversion[4], achieving high selectivity remains a critical challenge, because the formation of C$_{2+}$ products relies heavily on the stability of carbon-based intermediates at the catalytic active sites to enable efficient C–C coupling[5–7].

Among various developed photocatalysts for CO$_2$ reduction reaction (CO$_2$RR), transition metal sulfides, such as CdS, Bi$_2$S$_3$, CuInSnS$_4$, have attracted particular interest due to their tunable $d$-orbitals[8–10]. However, these materials often exhibit weak adsorption of

C$_1$ intermediates, e.g., *CO, leading to premature desorption[11–13] and preferential formation of C$_1$ products, such as CO, CH$_4$, and HCOOH, rather than the desired C$_{2+}$ compounds (Fig. 1a).

To address this limitation, several effective strategies, including heterojunction construction, crystal facet engineering, doping, and defect engineering, have been explored to enhance C$_{2+}$ selectivity by stabilizing carbon-based intermediates[14–17]. For instance, in MoS$_x$/Fe$_2$O$_3$ system, $d$-$p$ orbital hybridization at Mo-Fe sites reduced electrostatic repulsion between *CO and *COH intermediates, promoting ethylene (C$_2$H$_4$) formation[18]. Co-O-Fe triatomic sites in partially oxidized FeCoS$_2$ increased local charge density, enriching C$_1$ intermediates and directing the reaction pathway toward C$_2$ products[19]. Incorporation of Co into NiS$_2$ tailored the coordination number and

[1]State Key Laboratory of Heavy Oil Processing, Key Laboratory of Optical Detection Technology for Oil and Gas, China University of Petroleum, Beijing, China. [2]School of Chemistry and Chemical, In-situ Center for Physical Science, Shanghai Jiao Tong University, Shanghai, China. [3]School of Metallurgy and Environment, Central South University, Changsha, China. [4]State Key Laboratory of Powder Metallurgy, School of Physics, Central South University, Changsha, China. [5]Multidisciplinary Research Center, Institute of High Energy Physics, Chinese Academy of Sciences, Beijing, China. [6]University of Chinese Academy of Sciences, Chinese Academy of Sciences, Beijing, China. [7]School of Chemistry and Chemical Engineering, Ningxia University, Yinchuan, China. [8]These authors contributed equally: Zhiling Tang, Yingli Wang. ✉e-mail: weiyc@cup.edu.cn; minliu@csu.edu.cn; liuyunpeng@ihep.ac.cn; liuxi@sjtu.edu.cn

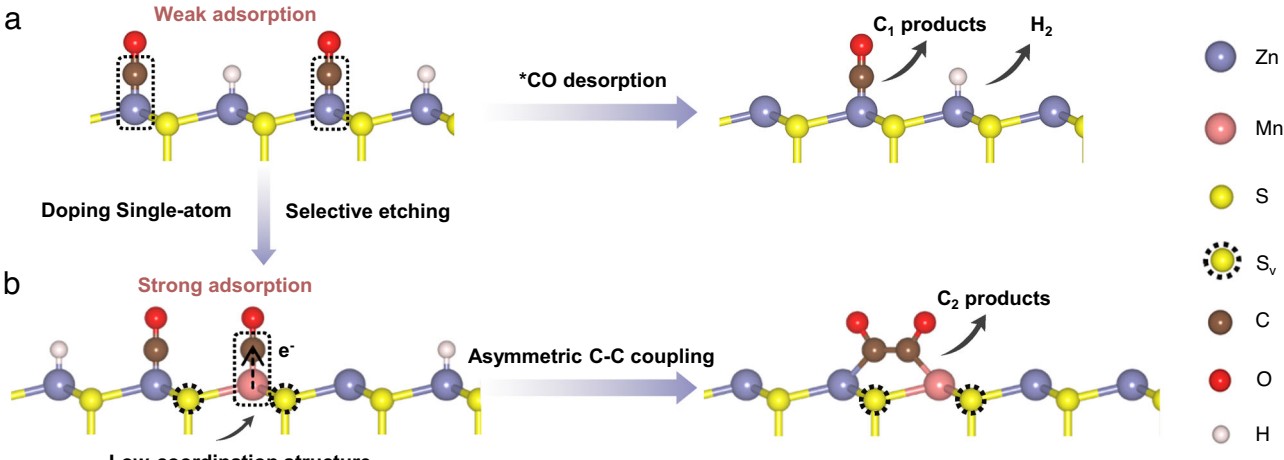

**Fig. 1 | Schematic diagram showing regulatory effect of coordination environment on *CO adsorption. a** On the pristine ZnS, the weak adsorption of *CO intermediates reduce the surface covering and then yields $C_1$ product. **b** The low-coordinated Mn single-atom sites enhance the adsorption of *CO intermediates via charge modulation, facilitating asymmetric C–C coupling.

oxide state of Ni sites, facilitating asymmetric C–C coupling between *CO species in atop and bridge adsorption configuration, achieving 75% selectivity for $C_2H_4$[20]. Despite these advances, achieving high activity and selectivity for photocatalytic $CO_2$-to-$C_{2+}$ conversion remains an unmet goal.

Single-atom catalysts (SACs), with atomically dispersed metal centers and well-defined coordination environments, provide a powerful platform for tuning intermediate adsorption and enhancing $CO_2RR$ activity and selectivity[18,21–24]. Particularly, low-coordination SACs can break geometric symmetry and modulate the local electronic structure, strengthening the adsorption of carbon-based intermediates and facilitating C–C coupling[25,26]. Therefore, integrating low-coordination SACs into binary sulfides holds great potential for stabilizing *CO and enabling efficient photocatalytic $CO_2$-to-$C_{2+}$ conversion (Fig. 1b).

Herein, we develop a low-coordination Mn single-atom embedded zinc sulfide ($Mn_1$–$ZnS_v$) via a microwave irradiation-induced targeted defect engineering strategy for enhancing $CO_2$-to-$C_2H_4$ photoreduction. Structural analyses reveal that Mn atoms are atomically dispersed and coordinated with a reduced number of sulfur atoms, forming Mn–Zn coupled sites with modulated electronic structures. Density functional theory (DFT) calculations and in-situ spectroscopic studies demonstrate that these low-coordination Mn sites effectively stabilize $C_2H_4$-related intermediates and promote asymmetric *CO–CHO coupling into *COCHO, a key intermediate for C–C bond formation. As a result, $Mn_1$–$ZnS_v$ exhibits attractive photocatalytic performance for $CO_2$ reduction, achieving a $C_2H_4$ production rate of 76.6 µmol g$^{-1}$ h$^{-1}$ with nearly 100% selectivity, significantly emphasizing the capability of regulating the coordination environment of sing-atom catalytic sites for photoconversion $CO_2$ into $C_{2+}$ products.

## Results
### Enhanced C–C coupling via low-coordination Mn sites
To assess *CO formation during the $CO_2RR$, we calculated the free energy profiles for $CO_2$-to-*CO conversion on ZnS, $Mn_1$–ZnS, and $Mn_1$–$ZnS_v$ (Fig. 2a, Supplementary Figs. 1–3, and Supplementary Data 1). Among these, $Mn_1$–$ZnS_v$ exhibits the lowest energy barrier (0.56 eV), significantly lower than that of ZnS and $Mn_1$–ZnS (Fig. 2b), indicating that low-coordination structure of Mn single-atom is favorable for $CO_2$ activation and *CO generation.

To investigate the behavioral of *CO intermediates on different coordination structures, the *CO adsorption energy and protonation energy barriers of $Mn_1$–ZnS and $Mn_1$–$ZnS_v$ were compared. The

interaction between the Mn $3d$ orbital and the $2\pi^*$ (LUMO) and $5\sigma$ (HOMO) orbitals of *CO underpins the modulation of adsorption behavior (Fig. 2c). The low-coordination Mn site in $Mn_1$–$ZnS_v$ strengthens the Mn-CO $\pi$ backbonding, resulting in a significantly enhanced *CO adsorption energy of −1.36 eV, much stronger than that of coordination-saturated $Mn_1$–ZnS (Fig. 2d). Furthermore, the incorporation of low-coordination Mn single atoms ingeniously balances the competitive adsorption of *CO intermediates and *H species (Supplementary Fig. 4). The appropriate binding inhibits premature *CO desorption and enables its participation in subsequent protonation steps. It is worth noting that the energy barrier for *CO hydrogenation on $Mn_1$–$ZnS_v$ is reduced to 0.75 eV, compared to 1.25 eV on $Mn_1$–ZnS, confirming the regulatory role of the low-coordination structure in steering the *CO reaction pathway (Supplementary Fig. 5).

To probe the origin of the stronger *CO adsorption, we investigated the electronic structure differences between $Mn_1$–ZnS and $Mn_1$–$ZnS_v$. Differential charge density analysis reveals a significant electron transfer (1.08 |e|) from the Mn single atom to the adsorbed *CO in $Mn_1$–$ZnS_v$ (Fig. 2e), compared to just 0.32 |e| in $Mn_1$–ZnS (Fig. 2f), highlighting the enhanced electronic interaction in the low-coordination structure. Furthermore, in $Mn_1$–$ZnS_v$ (coordination number, CN = 2), the Mn $3d$ orbital center shifts upward relative to $Mn_1$–ZnS (Fig. 2g), resulting in a higher $d$-band center (Fig. 2h). This upshift implies an increased population of anti-bonding states, favoring stronger adsorption of reaction intermediates and facilitating the key C–C coupling process.

To validate this mechanistic insight, we calculated the free energy barriers for C–C coupling through different $C_1$ species (*CO, *CHO, *COH) on $Mn_1$–$ZnS_v$ (Fig. 2i). The results suggested that direct dimerization of two *CO species into *COCO is energetically unfavorable (ΔG = 1.91 eV). Instead, *CO preferentially undergoes hydrogenation to *CHO, which subsequently couples with another *CO to form the *COCHO intermediate via an asymmetric C–C bond formation pathway (Supplementary Fig. 6). This pathway is thermodynamically more favorable and constitutes the lowest-energy pathway for $CO_2$-to-$C_2H_4$ conversion (Supplementary Figs. 7 and 8 and Supplementary Table 1). Thus, low-coordination Mn single-atom sites in $Mn_1$–$ZnS_v$ significantly enhance *CO adsorption and facilitates the asymmetric coupling of *CO and *CHO, ultimately leading to the selective generation of $C_2H_4$. The theoretical screening of coordination structures serves as a guide for establishing trends in *CO adsorption and the subsequent reaction pathways.

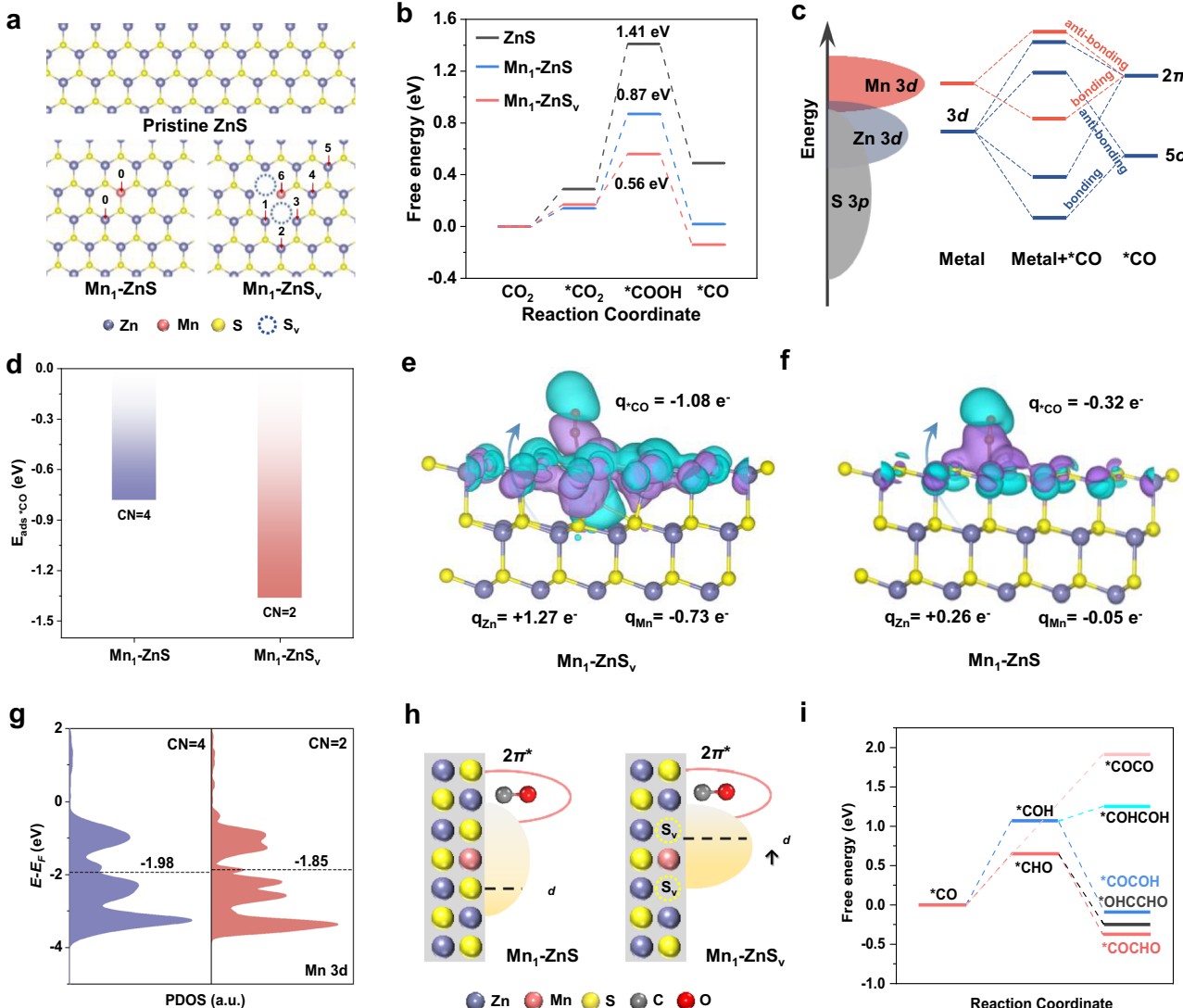

**Fig. 2 | Theoretical guidance. a** Schematic of calculation models for pristine ZnS, $Mn_1$–ZnS, and $Mn_1$–$ZnS_v$ (the atoms marked with numbers represent metal atoms in different coordination states). **b** The free energy diagram for $CO_2$ reduction reaction to display the conversion of $CO_2$-to-*CO over ZnS, $Mn_1$–ZnS and $Mn_1$–$ZnS_v$. **c** Illustration of the interaction between the $5\sigma$ and $2\pi^*$ of *CO and metal surfaces. **d** Adsorption energies of *CO on $Mn_1$–ZnS and $Mn_1$–$ZnS_v$. **e, f** Differential charge density and Bader charge of *CO adsorbed on $Mn_1$–$ZnS_v$ (**e**) and $Mn_1$–ZnS (**f**) (the isosurface is 0.005 e/Å$^3$). Blue and purple represent charge gaining and losing, respectively. **g, h** The partial density of states and the $d$-band center of the Mn atom for $Mn_1$–ZnS and $Mn_1$–$ZnS_v$, $\varepsilon_d$ represents the $d$-band center. **i** The free minimum-energy diagram for *CO hydrogenation and subsequent C–C coupling on $Mn_1$–$ZnS_v$ (111) facet. Source data are provided as a Source Data file.

## Catalyst synthesis and characterization

Guided by the DFT-screened $Mn_1$–$ZnS_v$ structure, a Mn single-atom substituted ZnS catalyst with targeted sulfur vacancies ($S_v$) was successfully synthesized via a microwave irradiation-induced targeted defect strategy (Supplementary Fig. 9), followed by $H_2O_2$ etching under an Ar atmosphere. Inductively coupled plasma optical emission spectrometry (ICP-OES) confirmed a Mn content of 0.6 wt% (Supplementary Table 2). X-ray diffraction (XRD) patterns (Supplementary Fig. 10) confirmed the formation of cubic-phase ZnS (PDF #05-0566)[27] with an average crystallite size of 41.3 ± 0.2 nm, as estimated using the Scherrer equation (Supplementary Fig. 11).

High-resolution transmission electron microscopy (HRTEM, Fig. 3a) revealed distinct lattice fringes with an interplanar spacing of 0.312 nm, corresponding to the (111) plane of ZnS. Scanning transmission electron microscopy–energy-dispersive X-ray spectroscopy (STEM-EDS) elemental mapping images (Fig. 3b) and the STEM-EDS line scanning profiles (Supplementary Fig. 12) corroborated the even distribution of Zn, Mn, and S throughout the sample, with no

observable aggregation of Mn into large clusters or particles. Aberration-corrected annular dark-field STEM (ADF-STEM, Fig. 3c) further confirmed the atomic dispersion of Mn, with isolated single atoms highlighted by yellow circles and localized intensity dips in the corresponding line-scan profile across a representative region.

To confirm the formation of $S_v$, electron paramagnetic resonance (EPR) spectroscopy was performed (Supplementary Fig. 13). A clear signal at $g = 2.005$ indicated the presence of $S_v$ in $Mn_1$–$ZnS_v$[28]. Additionally, EDS point analysis (Supplementary Fig. 14) showed an increased metal-to-sulfur atomic ratio, further supporting the generation of $S_v$, which increased with prolonged $H_2O_2$ etching time. Quantitative analysis from ICP-OES and EPR[29,30] (Supplementary Figs. 15 and 16 and Supplementary Table 2) revealed an isolated $S_v$ concentration of 1.54% for $Mn_1$–$ZnS_v$, closely matching the theoretically identified optimal configuration by DFT (Supplementary Fig. 17 and Supplementary Data 1).

To probe the local coordination environment of Zn and Mn atoms, X-ray absorption spectroscopy was conducted. X-ray absorption near-edge structure (XANES) spectra (Supplementary Fig. 18a)

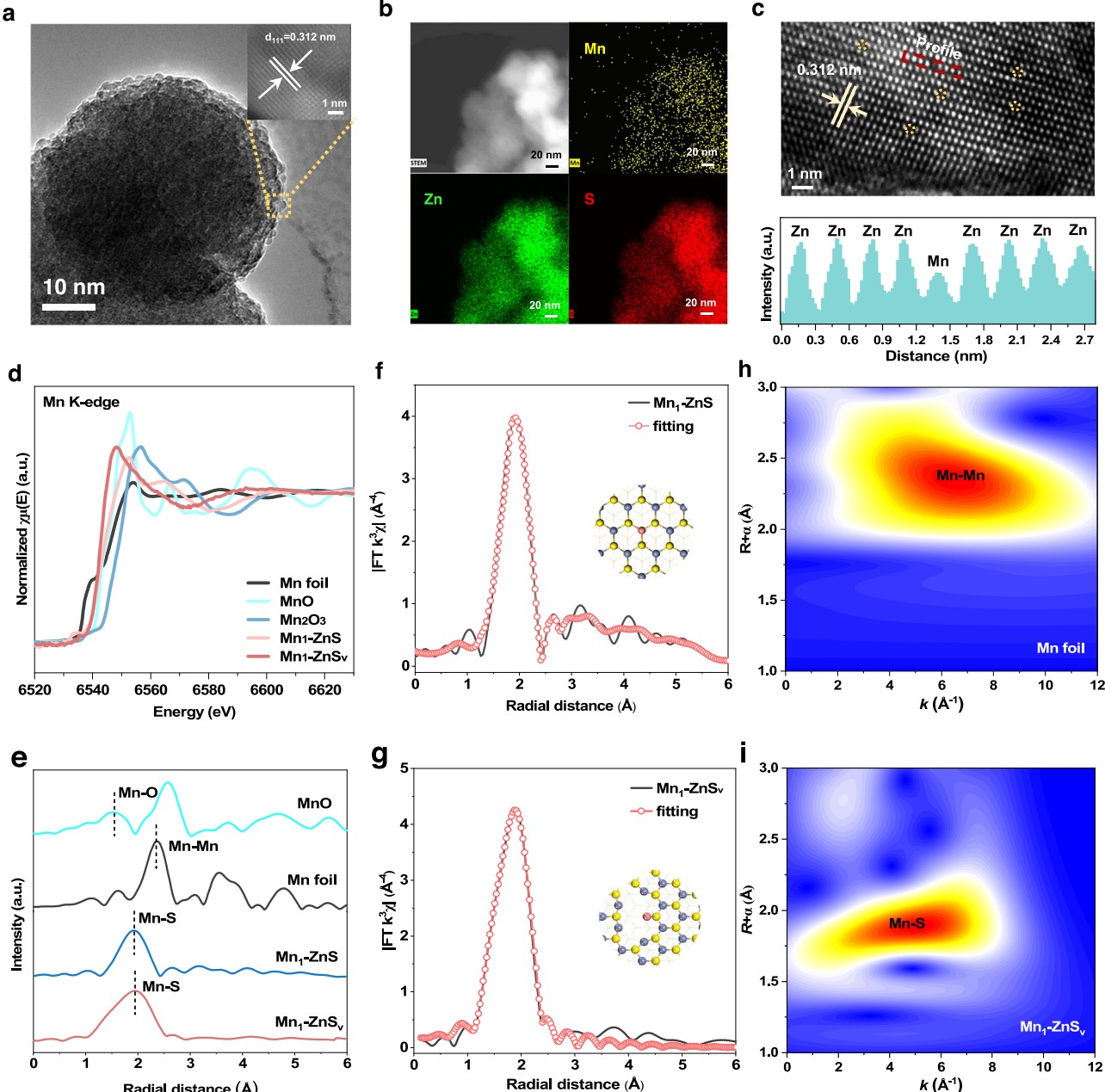

**Fig. 3 | Structural characterization. a, b** TEM image (**a**), STEM image and the corresponding EDS elemental mappings (**b**) of Mn$_1$–ZnS. Inset: An enlarged HRTEM of the dotted line area. **c** ADF-STEM image of Mn$_1$–ZnS$_v$ with corresponding intensity profile of the representative region (red dotted line), and the isolated single atoms highlighted with yellow circles. **d** Normalized Mn K-edge XANES spectra of Mn$_1$–ZnS, Mn$_1$–ZnS$_v$, and the related reference samples. **e** Fourier-transform (FT) spectra of EXAFS oscillations for Mn$_1$–ZnS and Mn$_1$–ZnS$_v$. **f, g** EXAFS spectrum fitting result of Mn$_1$–ZnS (**f**) and Mn$_1$–ZnS$_v$ (**g**) in R space. (The inserted image shows the atomic structure model of Mn$_1$–ZnS and Mn$_1$–ZnS$_v$, where the balls in blue, pink, and yellow represent Zn, Mn, and S atoms). **h, i** Wavelet transform (WT) contour plots of Mn foil (**h**) and Mn$_1$–ZnS$_v$ (**i**), respectively. Source data are provided as a Source Data file.

showed that Zn maintained an oxidation state of approximately +2.0. The Mn K-edge positions in both Mn$_1$–ZnS and Mn$_1$–ZnS$_v$ were located between those of metallic Mn and MnO, indicating an average Mn valence between 0 and +2 (Fig. 3d). Notably, Mn in Mn$_1$–ZnS$_v$ exhibited a lower oxidation state than in Mn$_1$–ZnS, attributed to the low-coordination environment induced by S$_v$ formation.

Fourier-transformed extended X-ray absorption fine structure (FT-EXAFS) analysis (Fig. 3e, Supplementary Fig. 18, and Supplementary Table 3) displayed distinct peaks corresponding to Zn–S and Zn–Zn coordination, while a prominent peak at 1.96 Å was assigned to Mn–S bonding. The absence of Mn–Mn and Mn–O coordination in both Mn$_1$–ZnS and Mn$_1$–ZnS$_v$ indicates that Mn atoms are atomically

dispersed and coordinated with neighboring S atoms, consistent with ADF-STEM results[31].

EXAFS fitting revealed that the first-shell Mn–S coordination number was 3.7 for Mn$_1$–ZnS and 1.9 for Mn$_1$–ZnS$_v$ (Fig. 3f, g and Supplementary Table 3). These values are in excellent agreement with DFT-optimized structures, confirming that Mn atoms in Mn$_1$–ZnS$_v$ are coordinated with only two sulfur atoms (insets in Fig. 3f, g). No Mn–Mn path in second-shell coordination (Supplementary Fig. 19) and wavelet transform (WT) analysis further confirmed the atomic dispersion and low-coordination environment of Mn in Mn$_1$–ZnS$_v$ (Fig. 3h, i).

To investigate the effect of low-coordination Mn single-atoms on light absorption efficiency, UV–Vis diffuse reflectance spectra and

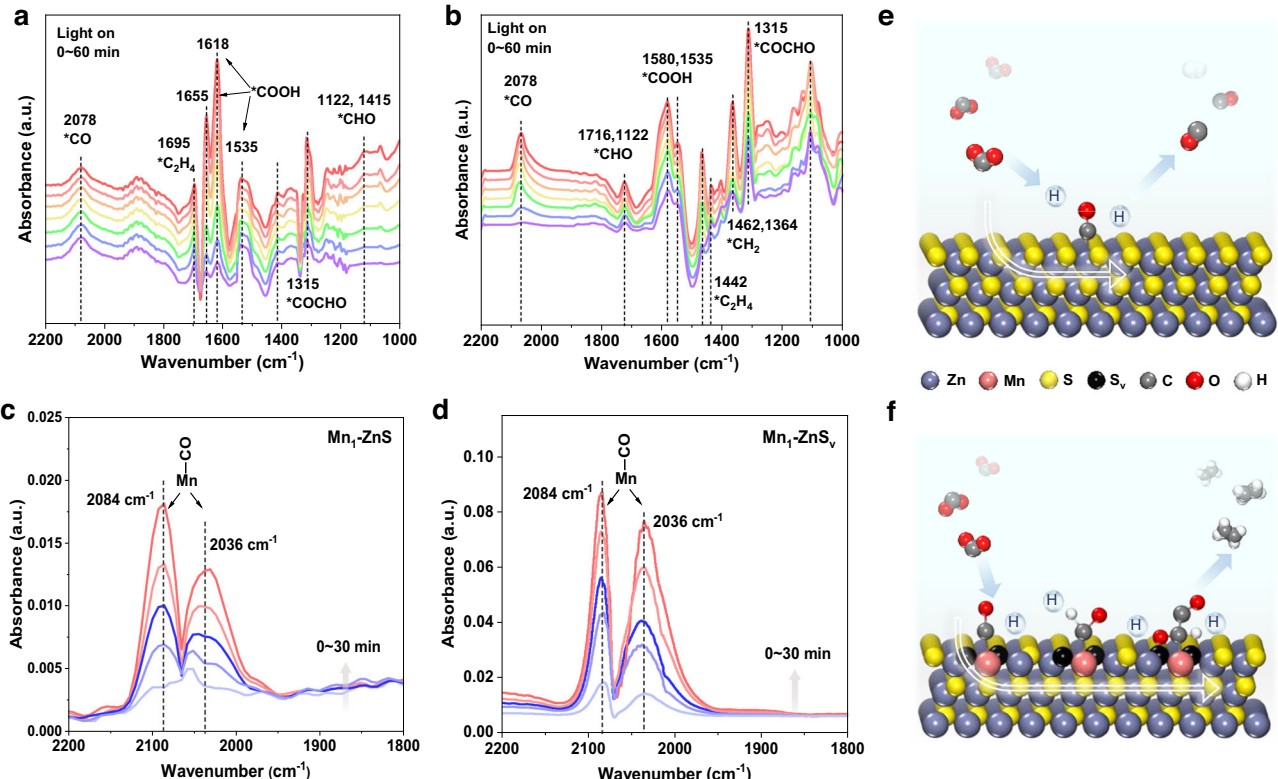

**Fig. 4 | In situ spectra analysis. a, b** In situ DRIFTS for $CO_2$ photoreduction reaction over $Mn_1$–ZnS (**a**) and $Mn_1$–$ZnS_v$ (**b**). **c, d** CO-adsorbed DRIFTS for $Mn_1$–ZnS (**c**) and $Mn_1$–$ZnS_v$ (**d**). **e, f** Reaction pathway schemes for ZnS and $Mn_1$–$ZnS_v$ interface. Source data are provided as a Source Data file.

X-ray spectroscopy (XPS) valence band spectroscopy were conducted. The optical absorption band edge of $Mn_1$–$ZnS_v$ is observed at 428 nm, corresponding to a smaller band gap of 2.88 eV (Supplementary Fig. 20). Ultraviolet photoelectron spectroscopy measurements revealed work functions ($\Phi$) of 6.13 and 5.60 eV for $Mn_1$–ZnS and $Mn_1$–$ZnS_v$, respectively (Supplementary Fig. 21). The valence band maxima for $Mn_1$–ZnS and $Mn_1$–$ZnS_v$ were located at 1.00 and 0.55 eV below the Fermi level (Supplementary Fig. 22), and the electronic band structure versus Normal Hydrogen Electrode could be elucidated (Supplementary Fig. 23), which is thermodynamically favorable for the $CO_2$-to-$C_2H_4$ conversion (Supplementary Fig. 24). These results are well-consistent with those obtained from Mott−Schottky plots (Supplementary Fig. 25), clearly suggesting the ability for $CO_2$-to-$C_2H_4$ conversion and $O_2$ evolution. Upon irradiation, photogenerated electrons are transferred from S (electron donor) to Mn (electron trap), and then rapidly injected into the adsorbed $CO_2$ molecules through the electron transfer channel (Supplementary Fig. 26). Moreover, improved charge-transfer dynamics induced by low-coordination Mn sites were recorded in Supplementary Figs. 27–29.

To further elucidate intermediate adsorption and reaction pathways in photocatalytic $CO_2$ reduction, in situ diffuse reflectance infrared Fourier transform spectroscopy (DRIFTS) was employed (Fig. 4a, b and Supplementary Table 4). Under simulated solar irradiation, $Mn_1$–ZnS exhibited characteristic signals at 1618, 1655, and 1695 cm$^{-1}$ (*COOH species)[26,32,33], 2078 cm$^{-1}$ (*CO)[34], 1122 and 1415 cm$^{-1}$ (*CHO)[33,35], and a key band at 1315 cm$^{-1}$ corresponding to *COCHO (Fig. 4a)[32], a critical intermediate in C−C coupling toward $C_2H_4$[36,37].

In contrast, $Mn_1$–$ZnS_v$ exhibited a stronger *CO signal at 2078 cm$^{-1}$ (Fig. 4b and Supplementary Fig. 30), indicating enhanced stabilization and coverage of *CO intermediate. The progressive intensification of the 1315 cm$^{-1}$ band over time confirmed efficient *CO−*CHO coupling and *COCHO formation, which was further verified by the $^{13}$CO labeling experiment (Supplementary Fig. 31). Additional bands at 1442, 1462,

and 1364 cm$^{-1}$ were attributed to *$C_2H_4$ and *$CH_2$ intermediates[32,38], confirming the high activity and selectivity of $Mn_1$–$ZnS_v$ toward $C_2H_4$ production.

To further probe CO adsorption behavior, CO-adsorbed DRIFTS was performed on both catalysts (Fig. 4c, d). Prominent peaks at ~2084 and 2036 cm$^{-1}$ were assigned to linearly bonded CO ($CO_L$) on Mn single atoms. $Mn_1$–$ZnS_v$ displayed significantly higher peak intensities than $Mn_1$–ZnS, indicating a stronger CO binding capability, consistent with DFT predictions. Based on the in situ DRIFTS data and theoretical insights, a photocatalytic reaction pathway for $CO_2$ conversion on $Mn_1$–$ZnS_v$ was proposed (Fig. 4e, f), highlighting the critical role of low-coordination Mn single-atom sites in stabilizing reaction intermediates, enabling *CO−*CHO coupling, and ultimately promoting selective $C_2H_4$ production.

## Evaluating catalyst performance for $CO_2$ photoreduction

The $CO_2$ photoreduction performance of $Mn_1$–$ZnS_v$ catalysts was assessed under visible light irradiation ($\lambda \geq 380$ nm) without the use of photosensitizers or sacrificial agents (Fig. 5, Supplementary Figs. 32–34, and Supplementary Table 5). As shown in Fig. 5a, pristine ZnS primarily produced CO at a rate of 64.6 µmol g$^{-1}$ h$^{-1}$, with a low $C_2H_4$ selectivity of only 5.6%. Compared to pristine ZnS, ZnS with sulfur vacancies ($ZnS_v$) exhibits negligible enhancement in $C_2H_4$ yield and selectivity (7.3%) (Supplementary Figs. 35 and 36). Upon incorporation of saturated-coordination Mn single atoms ($Mn_1$–ZnS), both activity and selectivity improved significantly, achieving a $C_2H_4$ formation rate of 47.5 µmol g$^{-1}$ h$^{-1}$ and a selectivity of 74.5%. Concurrently, CO evolution decreased to 28.4 µmol g$^{-1}$ h$^{-1}$ and the competing hydrogen evolution reaction was markedly suppressed (Supplementary Fig. 37), likely due to the *CO-stabilizing effect of Mn sites that promote its protonation and subsequent C−C coupling.

Notably, the low-coordination $Mn_1$–$ZnS_v$ catalyst delivered a significantly enhanced $C_2H_4$ formation rate of 76.6 µmol g$^{-1}$ h$^{-1}$, nearly

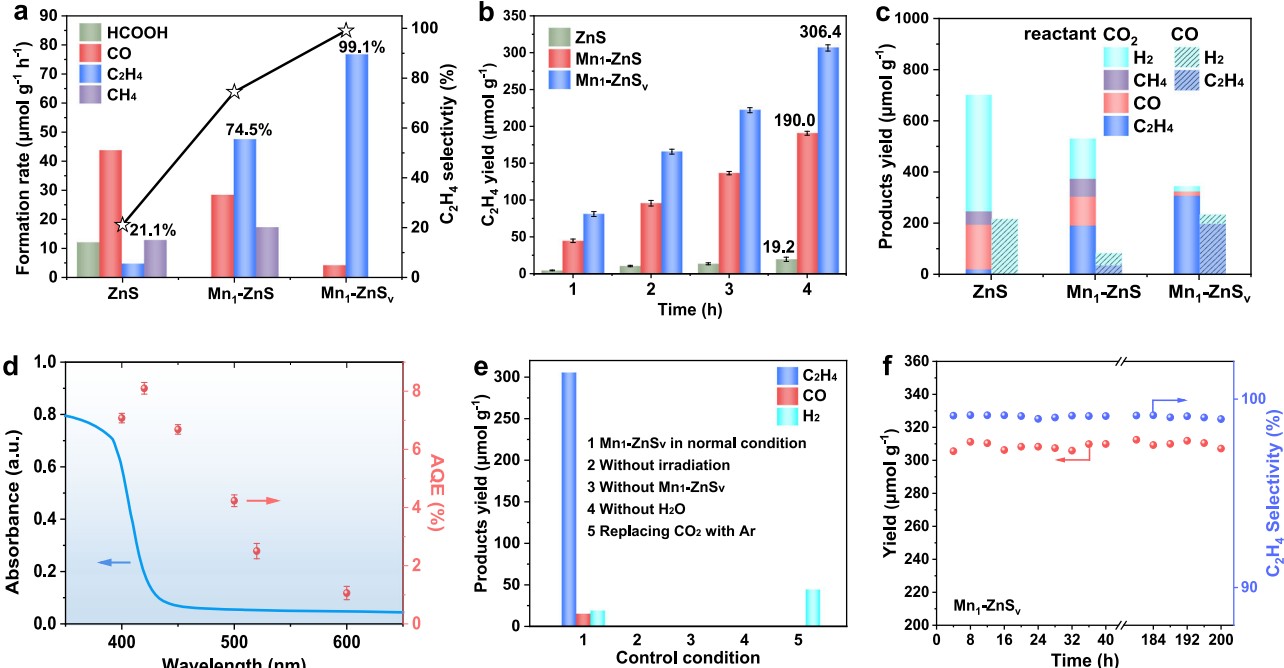

**Fig. 5 | Photocatalytic CO₂ reduction performance. a** Product yields of products (HCOOH, CO, C₂H₄, and CH₄) and the selectivity of C₂H₄ product over ZnS, Mn₁–ZnS, and Mn₁–ZnSᵥ catalysts (T = 298 K; irradiation time = 4 h; solvent: H₂O, 5 mL: amount of catalyst: 0.2 g). **b** The evolution of C₂H₄ product during photocatalytic CO₂ reduction over time (n = 3 independent experiments, data are presented as mean values ± SD). **c** The products yield over ZnS, Mn₁–ZnS and Mn₁–ZnSᵥ catalysts using CO₂ and CO as reactant, respectively. **d** The AQE of C₂H₄ evolution catalyzed by Mn₁–ZnSᵥ catalyst and the correlation between the related absorption spectra and absorption wavelengths (n = 3 independent experiments, data are presented as mean values ± SD). **e** The control experiments of photocatalytic CO₂ reduction performance over Mn₁–ZnSᵥ under altered conditions. **f** Cycling tests for CO₂ photoreduction to C₂H₄ over Mn₁–ZnSᵥ photocatalyst under the same as reaction conditions. Source data are provided as a Source Data file.

58.9 times higher than that of pristine ZnS, with a selectivity of 99.1% for C₂H₄ production (Fig. 5b and Supplementary Table 6). Meanwhile, CO formation further decreased to 4.2 μmol g⁻¹ h⁻¹ (Supplementary Fig. 37) and no liquid products were detected, underscoring the pivotal role of Mn coordination environment in modulating *CO adsorption and steering the reaction pathway toward selective C₂₊ hydrocarbon formation.

To further validate the role of low-coordination Mn SAs in promoting C–C coupling, a CO substitution experiment was carried out. This resulted in a C₂H₄ yield of 198.8 μmol g⁻¹ (Fig. 5c), confirming that *CO is a crucial intermediate for C–C coupling and effectively activating and stabilizing by the low-coordination Mn sites. The apparent quantum efficiency (AQE) for CO₂ photoreduction over Mn₁–ZnSᵥ reached 8.1% at 420 nm (Fig. 5d). Control experiments (Fig. 5e) demonstrated that C₂H₄ formation occurred only under light irradiation in the presence of both CO₂ and H₂O, excluding contributions from dark reactions or carbon-containing contaminants[39]. Isotope-labeling experiments using ¹³CO₂ and D₂O (Supplementary Fig. 38) further confirmed CO₂ and H₂O as the carbon and proton sources, respectively, for C₂H₄ formation. Considering that the activation of H₂O into active hydrogen species may be an important factor that affecting the turnover-limiting step during the photocatalytic CO₂RR to C₂H₄, the KIE of H/D value over the Mn₁–ZnSᵥ catalyst is close to 1 (Supplementary Fig. 39), further confirming the *CHO formation as the rate-determining step suggested by DFT results.

The photocatalytic stability of Mn₁–ZnSᵥ was evaluated over 50 consecutive cycles (200 h), during which no significant decline in activity or selectivity was observed (Fig. 5f). Post-reaction characterizations, including TEM, XRD, and XPS (Supplementary Fig. 40), confirmed the preservation of the structural and electronic integrity of the catalyst. No characterized signals of Mn–Mn bond over Mn₁–ZnSᵥ after long-term reaction further proved the great stability of the

coordination environment of Mn atoms (Supplementary Fig. 41), ensuring effective photocatalytic performance. Furthermore, the well-balanced charge consumption between the oxidation and reduction half-reactions ($R_{O/R} ≈ 1$) minimizes charge accumulation and effectively suppresses photocorrosion (Supplementary Table 5). Notably, under sacrificial-agent-free conditions, a comparative analysis of photocatalytic performance between our study and previous reported catalysts[17–19,32,40–50] is displayed in Supplementary Table 6. Both the competitive C₂H₄ yield and selectivity of Mn₁–ZnSᵥ underscores its potential as a highly efficient and robust CO₂-to-C₂H₄ conversion.

To evaluate the general applicability of low-coordination transition metal single atoms in ZnS matrices, analogous catalysts incorporating Fe, Co, Ni, and Cu were synthesized and tested (Supplementary Fig. 42). All M₁–ZnSᵥ (M = Fe, Co, Ni, Cu) variants exhibited substantially higher C₂H₄ yields than both pristine ZnS and their saturated-coordination M₁–ZnS counterparts, indicating a certain enhancement in photocatalytic performance induced by low-coordination metal centers (Supplementary Fig. 43).

To elucidate the electronic origin of this enhancement, projected density of states analyses were conducted for the 5σ and 2π orbitals of adsorbed CO and the metal 3d orbitals ($3d_{z^2}$, $3d_{xz}$, $3d_{yz}$) of the M₁–ZnSᵥ systems (Supplementary Fig. 44). Among all the systems, Mn₁–ZnSᵥ exhibits the highest d-band center at −1.85 eV, closer to the Fermi level. This upshift results in decreased population of antibonding d–5σ and bonding d–2π* states, thereby enhancing *CO adsorption.

Furthermore, integrated crystal orbital Hamilton population values for M–*CO bonds exhibited a clear linear decrease from Cu₁–ZnSᵥ to Mn₁–ZnSᵥ (Supplementary Fig. 45), further validating the stronger *CO binding affinity of Mn sites. In situ DRIFTS spectra (Supplementary Fig. 46) and free energy of reaction pathway (Supplementary Fig. 47) presented that Mn₁–ZnSᵥ exhibits significantly stronger *CO_L absorption peaks and comparatively favorable

energetics for *CO protonation and coupling than that of $Cu_1$–$ZnS_v$, which is consistent with the results from the catalytic performance. These findings conclusively demonstrate that low-coordination transition metal single atoms, particularly Mn, enhance *CO stabilization and facilitate the critical C–C coupling step, thereby driving selective $C_2$ product formation in $CO_2$ photoreduction.

## Discussion

In summary, we developed a $Mn_1$–$ZnS_v$ photocatalyst with a low-coordination Mn SA active site for efficiently selective $CO_2$ photoreduction to $C_2H_4$. The detailed studies revealed that low-coordination Mn single-atom active sites break the geometric symmetry of traditional SACs, resulting in anisotropic charge redistribution upon *CO adsorption. This not only enhances the adsorption of *CO intermediates, but also provides new sites for further C–C coupling. Both DFT calculations and experimental evidences confirmed the obtained $Mn_1$–$ZnS_v$ enhanced *CO adsorption and promoted the coupling between *CO and *CHO intermediates, ultimately driving the formation of $C_2H_4$. As a result, the optimal $Mn_1$–$ZnS_v$ photocatalyst achieved a $C_2H_4$ selectivity of 99.1% with a formation rate of 76.6 µmol g$^{-1}$ h$^{-1}$, and demonstrated long-term stability over 200 h. We anticipate that this strategy for tailoring low-coordination single-atom active sites will provide new insights into the rational design of highly efficient and selective photocatalysts for converting carbon dioxide into multicarbon products.

## Methods

### Chemical reagents

Zinc acetate ($Zn(OAc)_2$), Manganese acetate ($Mn(OAc)_2$), Copper acetate ($Cu(OAc)_2$), Iron acetate ($Fe(OAc)_3$), Nickel acetate ($Ni(OAc)_2$), Cobalt acetate ($Co(OAc)_2$), thiourea, ethylene glycol were purchased from Aladdin Reagent Co., Ltd., Shanghai. All chemicals are of analytical grade and used without further purification. The deionized water was supplied by a Millipore system (Outlet water resistivity > 18 MΩ cm).

### Sample preparation

To synthesize ZnS catalyst, typically 6 mmol of $Zn(OAc)_2$ and 12 mmol of thiourea were added to 100 mL of ethylene glycol. After continuous agitation for 30 min, the mixture was moved to a 250 ml three-necked flask with round bottom, and placed in a microwave reactor with a reflux unit. It was heated from room temperature to 150 °C with a heating rate of 25 °C min$^{-1}$ and maintained for 10 min. After the reaction cooled to room temperature naturally, the resulting powder was washed with deionized water and ethanol for three times and dried under vacuum at 80 °C for 6 h. The obtained samples were named as ZnS.

To synthesize $Mn_1$–ZnS catalyst, 6 mmol of $Zn(OAc)_2$ and 12 mmol of thiourea were added to 100 mL of ethylene glycol. After continuous agitation for 30 min, the different mass ratios of $Mn(OAc)_2$ (0.2%, 0.6%, 1%) was added to the suspension and continued stirring for 10 min. The mixture was moved to a 250 ml three-necked flask with round bottom, and placed in a microwave reactor with a reflux unit. It was heated from room temperature to 150 °C with a heating rate of 25 °C min$^{-1}$ and maintained in Ar flow for 10 min. After the reaction cooled to room temperature naturally, the resulting powder was washed with deionized water and ethanol for three times and dried under vacuum at 80 °C for 6 h. The obtained samples were named as $Mn_1$–ZnS-0.2, $Mn_1$–ZnS-0.6 and $Mn_1$–ZnS-1.0, respectively.

To synthesize $Mn_1$–$ZnS_v$, 100 mg $Mn_1$–ZnS was immersed in $H_2O_2$ solution (0.05 mol L$^{-1}$) under the protection of Ar in microwave reaction treated for 5, 10, and 15 s, respectively, all the obtained samples were carefully washed and dried using the method described above. The resulting samples were labeled as $Mn_1$–$ZnS_v$ with different S-vacancy concentrations.

The preparation process for $ZnS_v$ is the same as that of $Mn_1$–$ZnS_v$, except that the precursor Mn salt was not added. The synthesis steps for $Fe_1$–$ZnS_v$, $Co_1$–$ZnS_v$, $Ni_1$–$ZnS_v$, and $Cu_1$–$ZnS_v$ are the same as $Mn_1$–$ZnS_v$, except that the precursor salt was replaced with the corresponding acetate. Unless otherwise specified, the precursor transition metal salt loading for the ZnS sample was 0.6%.

### Characterization

The phase structure of the photocatalysts was performed on a powder X-ray diffractometer (XRD, Shimadzu XRD 6000). The morphology was observed via field emission scanning electron microscopy (FEI Quanta 200 F), transmission electron microscopy (TEM, JEOL JEM 2100). STEM-ADF images and EDS mapping were obtained by Hitachi HF5000, working at an accelerating voltage of 200 kV. To interpret the light absorption characteristics of the catalysts, UV–Vis DRS were detected on an UV–vis spectrophotometer (Shimadzu UV-2600) over a range of 200–800 nm. To study on the chemical bonds or functional information, in situ DRIFTS were examined using the IR Affinity-1 FTIR spectrometer. The Zn, Cu, and Mn K-edge X-ray adsorption spectra were acquired from 4B9A beamline in Beijing Synchrotron Radiation Facility (BSRF). Isotopic labelling experiments were performed on an Agilent 5977B GC/MS system, using $^{13}CO_2$ and $D_2O$ as reactants under the photocatalytic $CO_2$ reduction conditions. The GC-MS spectra reflect the relationship between the m/z of all ion fragments and their relative abundance. Photoluminescence spectra were analyzed on FluoroMax+ spectrophotometer (Hitachi, Japan) with excitation wavelength at 310 nm, and the receiving fluorescence range was from 400 to 700 nm. X-ray photoelectron spectroscopy (XPS) and valence band-XPS tests were conducted on the PerkinElmer PHI-1600 ESCA spectrometer with a Mg Kα X-ray source. All the calibration for the binding energies were based on the C 1 s at 284.8 eV[51]. Carbon dioxide-temperature programmed desorption ($CO_2$-TPD) was recorded on a BSD-C200 chemisorption system (BSD Instrument).

### In-situ DRIFT spectra measurement

In-situ DRIFT spectra were obtained from a Thermo Scientific Nicolet iS50 FTIR spectrometer. The compressed catalysts were placed in an diffuse reflectance cell (Harrick) equipped with a mixed atmosphere of $CO_2$ and $H_2O$ vapor for in situ experiments. In the pretreatment stage, the system was progressively heated from room temperature to 200 °C under continuous $N_2$ purging to remove surface impurities. After cooling to room temperature, the data were collected at a flow rate of 50 mL min$^{-1}$ in a mixed atmosphere. The infrared spectra obtained in the dark were used as background data. Subsequently, time-dependent IR spectra were collected under illumination for 30 min.

### In situ CO-adsorbed DRIFT spectroscopy measurement

In situ CO-adsorbed DRIFT spectroscopy measurement was carried out on a Nicolet 6700 instrument equipped with a mercury cadmium telluride detector. 10 mg catalyst was placed in an infrared diffuse reflection high-temperature reaction cell with a ZnSe window (Pike Technologies), pretreated in 10% $H_2/N_2$ flow at 300 °C for 30 min, then cooled to room temperature. This was followed by the introduction of CO flow for 30 min and subsequent evacuation. The CO-adsorbed FT-IR spectra were recorded at an apart of 2 min.

### Photoelectrochemical measurements

Photoelectrochemical (PEC) measurements were carried out in the three-electrode system by utilizing an electrochemical workstation (CHI660E). The prepared coated catalysts serve as the working electrode, with an effective surface area of 1 cm$^2$. Pt net and Ag/AgCl electrode were employed as the counter electrode and reference electrode, respectively. The $Na_2SO_4$ solution (0.1 mol L$^{-1}$) was acted as electrolyte (pH = 7.02 ± 0.05). In order to prepare the working electrode, 25 mg catalyst and 10 µL Nafion were dispersed in 384 µL deionized water and 96 µL ethanol, then followed by ultrasonication

for 25 min. The evenly mixed ink (5 μL) was dropped onto the surface of the glass-carbon electrode, and dried naturally at room temperature. A 50-s light-on/light-off cycle was applied to record the transient photocurrent response. The Mott−Schottky measurements were executed with the different frequencies (100, 500, and 1000 Hz), and electrochemical impedance spectra was performed, traversing a frequency range from 0.1 to 100000 Hz. Resistance (R) was measured to be $0.60 \pm 0.02\,\Omega$ at the open circuit potential.

### Tests on photocatalytic CO₂ reduction with H₂O

The photocatalytic $CO_2$ reduction tests were performed in a sealed reaction system by employing the Labsolar 6 A reactor (Beijing Perfectlight Technology Co., Ltd.). 0.2 g of photocatalysts was dispersed in 5 mL of deionized water within a glass dish (diameter is ~4.0 cm). The mixed sample was then placed inside a reactor (volume of ~370 mL) of the photoreaction system, at a distance of 15.0 cm from the light source. The reaction system was subjected to three cycles of vacuuming and refilling with high-purity $CO_2$, achieving a final $CO_2$ pressure of 80.0 kPa in the reactor. And a cooling circulating water device was utilized to maintain the reaction system a constant temperature at $25 \pm 1.5\,°C$. To simulate the natural photosynthesis, a 300 W xenon lamp (~100 mW cm⁻²) equipped with a 380 nm external filter served as light source. The light intensity of catalyst surface was measured by a ThorLabs PM100D Power with a photodiode sensor, with the average value determined from multiple regions of the watch glass[52]. After starting the light source and the gas chromatograph automatic sampling device, the sample was analyzed every 30 min, and the reaction test was carried out for 4 h. The amount of $C_2H_4$, $CH_4$, $CO$, $H_2$, and $O_2$ products evolved was analyzed online by a GC-9560 gas chromatograph (HuaAiSePu Company) using the external standard calibration method. For the liquid product, the amount of HCOOH was determined by a proton nuclear magnetic resonance (¹H NMR) analysis, taking DMSO as an internal standard. No other products were detected above the detection limit of instrument.

The product selectivity for $CO_2$ photoreduction to $C_2H_4$ was calculated according to the following equations:

$$S_{C_2H_4}(\%) = \frac{12n(C_2H_4)}{12n(C_2H_4) + 2n(CO) + 8n(CH_4) + 2n(CHOOH)} \times 100\% \quad (1)$$

where $n$(product) is the molar amount of product generated.

To quantify the efficiency of photocatalytic light-energy conversion, the apparent quantum efficiency (AQE) was measured. The incident photon numbers were determined using monochromatic light sources at specified wavelengths ($\lambda$ = 400, 420, 450, 500, 520, and 600 nm). AQE was defined as the ratio of consumed photons to incident photons, and according to the following equations:

$$AQE_{C_2H_4} = \frac{12 \times \text{Number of reacted electrons}}{\text{Number of incident photons}} \times 100\%$$
$$= \frac{12 \times n_{C_2H_4} \times N_A}{S \times I \times t \times \frac{\lambda}{h \times c}} \times 100\% \quad (2)$$

where $S$ is the irradiated area (cm²), $I$ is the irradiation light intensity (W cm⁻²), $t$ is the irradiation time (s), $\lambda$ is the equivalent wavelength (m), $h$ is Planck's constant $(6.626 \times 10^{-34}\,J\,s^{-1})$, $c$ is the speed of light $(2.998 \times 10^8\,m\,s^{-1})$ and $N_A$ is Avogadro's number $(6.022 \times 10^{23}\,J\,mol^{-1})$. $n_{C2H4}$ is the amount of $C_2H_4$ produced in 4 h.

The incident photon-to-current efficiency is defined as the ratio of the incident monochromatic photons converted to collected electrons

and can be calculated by using Eq. (3):

$$IPCE = \frac{\frac{hc}{\lambda} \times j}{Ie} \quad (3)$$

where $j$ is the photocurrent density (A cm⁻²), $\lambda$ is the equivalent wavelength (m), and $I$ is the irradiation light intensity (W cm⁻²).

### Density functional theory (DFT) calculations

DFT calculations were carried out in the framework of the Vienna ab initio Simulation Package (VASP). The exchange and correlation energies were established by the Perdew−Burke−Ernzerhof functional with spin-polarized generalized gradient approximation[53]. The core electrons were described by the projected augmented wave pseudopotentials. For geometry optimization, a cutoff energy of 450 eV was used to expand the wave functions, and the Brillouin zone was sampled with $2 \times 2 \times 1$ k-points. All the structures and energy were allowed to relax below $0.05\,eV\,Å^{-1}$. The DFT-D3 method of Grimme et al. was employed to involve the van der Waals interaction[54]. The calculations of charge density difference were employed to analyze the movement and distribution of the charge. For the elementary reaction barriers, the transition states were determined by the climbing image nudged elastic band method and were confirmed by further frequency calculations showing one and only one imaginary frequency.

According to the XRD results, a $5 \times 5$ slab model composed of three layers along the (111) direction was constructed with the space group F-43m model of ZnS carrier. A 20.0 Å vacuum region between the slabs was constructed to avoid the interlayer interaction. The doped ZnS with other transition metal elements was simulated by replacing one of the Zn atoms. The S-vacancy was simulated by removing S atoms off, and the theoretical value of $S_v$ is 1.33%. Considering the difference in the performance of as-prepared catalysts, ICP quantified vacancy concentrations, specific configuration of sulfur vacancies was adopted to model the surface. The atomic coordinates of the DFT models are provided as a Supplementary Data 1 file. The formation energy of sulfur vacancy $(\Delta E_{VS})$ was calculated as follows:

$$\Delta E_{VS} = E_{slab-V_S} - E_{slab} - E_S \quad (4)$$

in which $E_{slab-V_S}$ and $E_{slab}$ represent the energy of the slab after and before $V_s$. The adsorption energy $E_{ads}$ was calculated by a standard formula:

$$E_{ads} = E_{catalyst\text{-}*} - E_{catalyst} - E* \quad (5)$$

in which $E_{catalyst\text{-}*}$ and $E_{catalyst}$ is the total energy of the slab with and without intermediates.

The Gibbs free energy change ($\Delta G$) for the reaction was calculated by

$$\Delta G = \Delta E + \Delta E_{ZPE} - T\triangle S \quad (6)$$

in which $\triangle E$ is reaction energy obtained from DFT calculations, $\triangle E_{ZPE}$ represent the change of the zero-point energy, T$\Delta$S is the entropic contribution (T was set to be 300 K), which were obtained from the vibrational frequency calculations through the VASPKIT code. The free energy of the proton−electron pair is equal to half the free energy of the hydrogen molecule according to the computational hydrogen electrode method[55].

Ab initio molecular dynamics simulations are carried out by CP2K package and the QUICKSTEP module with fully explicit solvent water molecules[56]. Godecker-Teter-Hutter (GTH) pseudopotentials were adopted to model the core electrons, and double-ζ valence single polarization (DZVP)-molecularly optimized (MOLOPT)-short-ranged-GTH basis set was employed to obtain optimized structure[57,58]. Valence electrons were

expanded in an orthonormal plane-wave basis using a cutoff energy of 400 Ry, along with Grimme D3 dispersion correction[59]. The self-consistent field convergence criterion is set to $10^{-5}$ Ry for the total energies, and the simulation temperature (300 K) was controlled through a Nosé-Hoover thermostats with time step of 1.0 fs. DFT offers valuable insights for screening coordination structures and exploring reaction pathways, yet it remains inherent limitations in simulating practical catalytic conditions (e.g., actual illumination, solvation effects). Thus, the computationally identified coordination structures should serve as qualitative guides for catalyst design and mechanistic investigation.

## Data availability

All data that support the findings of this study are present in the paper and the Supplementary Information. Source data are provided with this paper.

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

## Acknowledgements

This work was supported by the National Natural Science Foundation of China (22376217 to Y.C.W., 22376222 to M.L., and 12305372 to Y.P.L.), the National Key Research and Development Program of China (2022YFB3504100 to Y.C.W. and 2024YFC3712104 to M.L.), the Carbon Neutrality Research Institute Fund of Shandong Institute of Petroleum and Chemical Technology (CNIF20240106 to Y.C.W.), the Science and Technology Innovation Program of Hunan Province (2023RC1012 to M.L.), the Central South University Research Program of Advanced Interdisciplinary Studies (2023QYJC012 to M.L.). The authors wish to thank facility support of the 4B9A beamline of Beijing Synchrotron Radiation Facility (BSRF).

## Author contributions

Y.W., M.L., and Y.L. conceived the project and directed the project. Z.T. carried out the experiments, analysed the experimental data, performed DFT calculations, and wrote the manuscript. Y.W. (Yingli Wang) synthesized the materials and performed the photocatalytic activity tests and in-situ experiments. T.Q. and X.L. performed the ADF-STEM experiments. J.X., X.W., X.L. (Xuanzhen Li), and Y.L. performed the XANES and EXAFS measurements. Y.W. and Z.Z. proposed the project. Y.W. and M.L. supervised the project, discussed the data, and revised the manuscript.

## Competing interests

The authors declare no competing interests.
