## [Transparent Peer Review file · Nature Communications]

Near-unity CO₂-to-ethylene photoconversion over low coordination single-atom catalysts

Corresponding Author: Professor Min Liu

Version 0:

Reviewer comments:

Reviewer #1

(Remarks to the Author)

This manuscript presents an interesting study where atomic-level coordination control in a sulfide host translates directly to near-unity C₂ selectivity in sacrificial-agent-free CO₂ photoreduction. The key contribution of this work is the deliberate creation of sulfur vacancies around the Mn center, lowering its coordination from the usual tetrahedral (CN = 4) to CN ≈ 2. This unsymmetrical Mn–S₂ motif is argued to (i) strengthen *CO adsorption, (ii) lower the *CO → *CHO barrier, and (iii) enable *CO–*CHO asymmetric coupling to *COCHO, the gateway intermediate for C–C coupling. The DFT-guided synthesis, operando DRIFTS, XAFS, AC-STEM, and isotope (¹³CO₂/D₂O) labeling are combined to provide an unusually complete mechanistic picture for a photocatalytic CO₂RR study. As such, I would like to recommend to publish this work with major revisions.

1. EPR (g = 2.005) and S-content from EDS indicate Sv formation, yet the absolute Sv concentration (1.54 %) is derived from a spin-counting model that assumes isolated S-vacancies. Clustering of vacancies could alter Mn coordination and electronic structure; spatially resolved EELS or positron annihilation spectroscopy would help to get S vacancy quantification and homogeneity.
2. Only 0.6 wt % Mn is loaded; at this level, even a small fraction of MnOx or MnS₂ clusters (<1 nm) could be missed by AC-STEM. A Mn K-edge EXAFS fitting range beyond 3 Å should be inspected for Mn–Mn contributions.
3. Photocurrent and EIS (Supplementary Figs. 18–19) show improved charge-transfer kinetics for Mn₁–ZnS_v, but the incident-photon-to-current efficiency (IPCE) at 420 nm is not reported; this would directly corroborate the 8.1 % AQE claim.
4. Mott–Schottky plots give flat-band potentials, yet the absolute band positions (vs. NHE) are extracted assuming no Fermi-level pinning by Mn states; in situ ambient-pressure XPS under illumination could verify band alignment under operating conditions.
5. D₂O experiments confirm water as the proton donor, but no KIE (k_H/k_D) for C₂H₄ formation is provided. A primary KIE would strengthen the claim that *CHO formation is turnover-limiting.
6. The H₂ evolution rate is suppressed to ≈1 μmol g⁻¹ h⁻¹, but its potential role as a competing pathway is not quantitatively modeled.
7. Ten 4-h cycles (40 h total) show no loss, yet photocorrosion of ZnS in aqueous media is well-known. Longer-term tests (>100 h) or continuous-flow operation should be addressed.
8. Microwave/H₂O₂ etching is elegant for lab scale (~100 mg batches); scalability to gram or kilogram scale is not addressed.
9. Only the ZnS(111) facet is modeled while ZnS also exposes (110) and (100) surfaces with different Sv formation energies. It will be necessary to model other two surfaces using DFT calculations for comparison.
10. Solvation and entropic contributions to *COCHO stability are not explicitly included.
11. ab-initio molecular dynamics simulations should be performed to capture dynamic vacancy reconstruction (e.g. 300 K).
12. Band-gap determination from UV–Vis (Fig. S13) uses a Tauc plot without stating whether direct or indirect transition is assumed; please clarify.
13. The reaction chamber volume, light intensity mapping, and temperature rise under focused Xe lamp are not documented; local heating can accelerate C–C coupling.
14. DRIFTS spectra are normalized to the ZnS phonon band at 1122 cm⁻¹, but absolute absorbance calibration would allow estimation of surface *CO coverage.
15. The error bars for AQE (currently single value at 420 nm) should be provided.

(Remarks on code availability)

Reviewer #2

(Remarks to the Author)

The manuscript addresses a significant challenge for the energy community – how to efficiently convert CO₂ into valuable C₂ products using solar energy directly. The manuscript is well-written, clearly organized, and contains a wide spectrum of characterization methods. My review will focus primarily on the photochemical characterization experiments, which require some clarifications and the filling of missing data points to assess the paper's merit accurately.

1. The paper's main point is that using low-coordination Mn sites to stabilize CO₂R intermediates results in a higher efficiency towards C₂+ products, as otherwise these critical intermediates are not able to react further. To support this claim, the authors should also perform control experiments using CO as feed for the ZnS catalyst and add this to Fig 5c. This is a critical point to address.
2. Secondly, the authors claim that they excelled in two metrics important for photochemical systems: ethylene production rate and ethylene selectivity. Given this is a primary claim, the authors should provide more details on how the experiments were conducted and how the metrics were calculated. Also, were the experiments repeated for reproducibility? The error bars should be added to the photochem experiments. The authors should also include their calculations of apparent quantum efficiency (specific values used, not only the formula).
3. Important details are missing in the experimental description -e.g., what was the total amount of CO₂ injected into the reactor, and how was the photocatalyst immobilized?
4. Fig 5d: control condition 5, should it be "replacing CO₂ with Ar" instead of "replacing Ar with CO₂"?
5. What do authors mean as "efficiency of tourism" (SI, section on Tests on photocatalytic CO₂ reduction with H₂O)? there is also a typo in Eq1. Looks like this was an auto-correct mistake, please review all your writing carefully.

(Remarks on code availability)

no code

Reviewer #3

(Remarks to the Author)

(Remarks on code availability)

Reviewer #4

(Remarks to the Author)

Tang et al. report a Mn single-atom catalyst embedded in ZnS with sulfur vacancies (Mn¹-ZnS_v) for photocatalytic CO₂ conversion to C₂H₄. Photocatalysts were modified by inclusion of Mn, hypothesizing that "single-atom" sites with Mn would increase binding energy for *CO and steer selectivity to C₂ products, Figure 1. Nanoparticle catalysts were made by solution-based method: ZnS and Mn-ZnS with 3 different mass ratios. Fe, Co, Ni, and Cu were also employed as dopants. S vacancies were induced by microwave treatment for 5-15 seconds: Zn-ZnS_v series. While a large number of samples appear to have been made, study focuses on one of them (page 6): "Inductively coupled plasma optical emission spectrometry (ICP-OES) confirmed a Mn content of 0.6 wt% (Supplementary Table 1)." It does appear, EPR in Figure S8, that this sample has more unpaired electrons than control which did not undergo the microwave treatment. A very high selectivity for this preparations is claimed: 76.6 μmol·g⁻¹·h⁻¹ under sacrificial-agent-free conditions. No sacrificial species were employed and ¹³C and ²D labeling was used to ensure that carbon and protons come from CO₂ and water, respectively. O₂ production is also reported, Figure 22. Study thus follows "best practices" for experimental reports of this type, see 10.1021/acseenergylett.8b00196 and 10.1002/anie.201207199. Also, interestingly, Mn-ZnS_v is able to convert CO to C₂H₄ (this is not always the case for CO₂R photocatalysts). They estimate an AQE of 8% at 420 nm. The authors attribute the performance to *CO stabilization and *CO-*CHO coupling, supported by in situ spectroscopy and DFT calculations.

Review prompts are addressed as follows:

What are the noteworthy results?

The performance of the photocatalysts, both in terms of activity/selectivity, is very good compared to other reports in the literature, Figure 5f.

Will the work be of significance to the field and related fields? How does it compare to the established literature? If the work is not original, please provide relevant references.

While the performance metrics are high, the novelty is limited. Similar catalyst architectures and comparable activity/selectivity have been reported (e.g., Energy Environ. Sci. 17, 5060–5069 (2024); J. Am. Chem. Soc. 145, 422–435 (2023)), and the present study does not clearly distinguish itself conceptually or mechanistically.

Does the work support the conclusions and claims, or is additional evidence needed?

In part, see detailed comments below.

Are there any flaws in the data analysis, interpretation and conclusions? Do these prohibit publication or require revision?

Revisions are needed in the data analysis, see detailed comments below.

Is the methodology sound? Does the work meet the expected standards in your field?

Yes. Again, the use of isotope labeling, CO substitution, etc. are best practices in the field.

Is there enough detail provided in the methods for the work to be reproduced?

Yes.

Summary statement.

The manuscript reports strong catalytic performance but lacks sufficient novelty and mechanistic depth to meet the standards of Nature Communications. Addressing the above points, particularly clarifying the Mn–vacancy interplay, rigorously validating the universality claim, and strengthening stability and mechanistic evidence, would be essential to enhance the originality and impact of the work.

Detailed comments.

The interplay between sulfur vacancies and Mn single atoms is insufficiently resolved. Figure S26 presents 4 h C₂H₄ yields for Fe, Co, Ni, and Cu-doped catalysts subjected to the microwave treatment. Yields are ~70-120 $\mu\text{mol/g}$. What is not shown on this graph is the Mn-ZnS not subjected to the microwave treatment: 190 $\mu\text{mol/g}$. This data does not support the statement: “indicating a universal enhancement in photocatalytic performance induced by low-coordination metal centers.” So it is not clear whether Mn substitution, which appears to be the original hypothesis, or the S vacancy is the main effect. At the very least, a ZnS_v control (without Mn) should be synthesized, characterized, tested, and included in DFT calculations. Figures 4a and 4b do not convincingly isolate the effect of sulfur vacancies. Full catalytic cycle calculations with corresponding energy profiles are needed for both cases, and any synergistic effect should be explicitly demonstrated in the mechanism.

The assertion that multiple metals (Fe, Co, Ni, Cu) can be incorporated as single sites requires direct evidence. At least one alternative metal (e.g., Cu) should be systematically compared with Mn via in situ DRIFTS, XAS, and full DFT energy profiles, etc., with a deeper discussion of performance differences beyond the d-band center argument.

Figure 2. In the free energy diagram, conversion of CO₂(g) to *CO is shown to have a negative free energy (in principle, spontaneous). What conditions were assumed to make this reaction downhill. I suspect that the authors are using the computational hydrogen electrode that the reaction is proceeding via proton-coupled electron transfers. If this is the case, what potential was assumed and why was it thought that this could be realized in photocatalytic experiments?

More generally, the influence of light on the catalytic process, particularly from a mechanistic perspective, is not discussed in sufficient detail. DFT calculations, especially those related to Supplementary Fig. 3, should incorporate the effect of photoexcitation to better reflect the actual reaction conditions.

Supplementary Fig. 25 is ambiguous. Given the claimed 200-hour stability (Fig. 5e), post-reaction characterizations, especially XAS, should be shown for the spent catalyst. More frequent GC sampling (e.g., every 5–10 min) is recommended to capture potential fluctuations.

¹³C labeling, combined with in situ DRIFTS, would provide stronger evidence for the proposed asymmetric *CO–*CHO coupling pathway.

Figure S24. This is said to be a GC-MS spectrum. But multiple analytes are present. Were chromatograms summed to produce this data?

The following is said about Figure 3b. “Elemental mapping via energy-dispersive X-ray spectroscopy (EDS, Fig. 3b) showed homogeneous distribution of Zn, Mn, and S throughout the sample, with Mn atoms clearly penetrating into the ZnS matrix.” On what basis can it be said that the Mn distribution is homogeneous? What would an inhomogeneous distribution look like? Also “Mn atoms clearly penetrating into the ZnS matrix.” It is not possible to see if Mn atoms are inside the ZnS with TEM as it produces a 2D projection of a 3D object.

Figure 3c caption. What do the yellow circles mean and what quantitative criterion was used to draw them?

Page 6. “with directional sulfur vacancies (S_v).” What is a directional sulfur vacancy? ZnS has the zinc-blende structure, Figure 6. There is only one type of S in this structure and just one type of S vacancy. Also, how can a vacancy have a direction (vectorial?).

Page 7. “...revealed a S_v concentration of 1.54% ¹⁵⁶Mn₁–ZnS_v, closely matching the theoretical value predicted by DFT.” How can DFT predict the vacancy concentration produced by the microwave treatment?

Page 9. "In contrast, Mn₁-ZnS_v exhibited a stronger *CO signal at 2078 cm⁻¹ (Fig. 4b), indicating enhanced generation and stabilization of *CO." Authors cannot conclude that CO₂ → CO is faster on the Mn-doped material. The steady-state concentration of *CO depends both on the generation and desorption rate.

(Remarks on code availability)

Reviewer #5

(Remarks to the Author)

See attached file

(Remarks on code availability)

Version 1:

Reviewer comments:

Reviewer #1

(Remarks to the Author)

Following a comprehensive re-evaluation of the manuscript, it has been concluded that the method for free energy calculation remains flawed. Specifically, the use of the electrochemical potential (U) in equation 5 is inappropriate, as it represents the conduction band potential of the semiconductor photocatalyst, which is not applicable in this context. Furthermore, in equation 3, the energy should correspond to the entire catalyst surface slab rather than just the surface. Additionally, the manuscript does not provide sufficient details regarding the calculation of the free energy barrier. In light of these significant issues, I regret that I cannot recommend this work for publication in its current form.

Reviewer #2

(Remarks to the Author)

The authors addressed my comments.

Reviewer #3

(Remarks to the Author)

Reviewer #4

(Remarks to the Author)

Revised ms. which contains new experimental data and clarifies many points is recommended for publication.

Reviewer #5

(Remarks to the Author)

The authors have already provided reasonable answers to all the questions that were raised.

Version 2:

Reviewer comments:

Reviewer #1

(Remarks to the Author)

The authors have provided reasonable replies to all my queries and have appropriately revised the manuscript accordingly. Now I do not have any further questions regarding the manuscript.

Response to Reviewers' Comments

Manuscript number: NCOMMS-25-62587A

Title: Near-unity CO₂-to-ethylene photoconversion over low-coordination manganese single-atom sites on binary sulfide

A Point-to-point Response to Reviewer's comments

Dear Reviewers,

Thank you for checking our manuscript and proposing valuable comments! The comments are highly appreciated and helpful to improve our work. We have made corresponding changes to the manuscript (marked in RED) and Supplementary information to address the reviewers' concerns; these changes are specified and discussed in a point-to-point response to the reviewers' comments, as shown below:

REVIEWER COMMENTS

Reviewer #1 (Remarks to the Author):

This manuscript presents an interesting study where atomic-level coordination control in a sulfide host translates directly to near-unity C₂ selectivity in sacrificial-agent-free CO₂ photoreduction. The key contribution of this work is the deliberate creation of sulfur vacancies around the Mn center, lowering its coordination from the usual tetrahedral (CN = 4) to CN \approx 2. This unsymmetrical Mn–S₂ motif is argued to (i) strengthen *CO adsorption, (ii) lower the *CO \rightarrow *CHO barrier, and (iii) enable *CO–*CHO asymmetric coupling to *COCHO, the gateway intermediate for C–C coupling. The DFT-guided synthesis, operando DRIFTS, XAFS, AC-STEM, and isotope (¹³CO₂/D₂O) labeling are combined to provide an unusually complete

mechanistic picture for a photocatalytic CO₂RR study. As such, I would like to recommend to publish this work with major revisions.

Response: We appreciate the reviewer's detailed summary of our work and are pleased that the novelty and strengths of our coordination design and its photocatalytic performance were well recognized. We have carefully addressed each of the reviewer's concerns below.

1. EPR ($g = 2.005$) and S-content from EDS indicate S_v formation, yet the absolute S_v concentration (1.54 %) is derived from a spin-counting model that assumes isolated S-vacancies. Clustering of vacancies could alter Mn coordination and electronic structure; spatially resolved EELS or positron annihilation spectroscopy would help to get S vacancy quantification and homogeneity.

Response: We sincerely thank the reviewer for this insightful and professional comment. We agree that clarifying the distribution of S-vacancies is crucial for confirming the precise coordination environment of the Mn center. We have addressed this concern through additional experiments and deeper analysis as detailed below.

First, from a theoretical perspective, a facile and thermally activated hydrogen peroxide (H₂O₂) chemical etching approach is proposed to introduce homogeneously distributed single S-vacancies onto the Mn₁-ZnS surface. Then, the systematic adjustment of the etching duration and etching temperature contributes to precise tuning of the single S-vacancy concentration. Based on the DFT calculations, Mn₁-ZnS_v means that two single S atoms in the pristine Mn₁-ZnS are removed, resulting in a theoretical S-vacancy concentration of 1.33% (a total of 75 S atoms in the pristine Mn₁-ZnS model). The structure model and electron localization function (ELF) of ZnS, Mn₁-ZnS and Mn₁-ZnS_v visually demonstrate the existence of S vacancy (**Fig. R1**).

Second, to confirm the formation of S_v , electron paramagnetic resonance (EPR) spectroscopy was performed (**Fig. R2a**). A clear signal at $g = 2.005$ indicated the presence of S_v in $Mn_{1-x}Zn_xS_v$. The narrow and symmetric line shape of our EPR signal ($g=2.005$) suggests negligible dipole-dipole interaction between the unpaired electrons, consistent with isolated defects (*J. Am. Chem. Soc.* **2020**, 142, 9, 4298–4308). Additionally, EDS point analysis (**Fig. R2b**) showed an increased metal-to-sulfur atomic ratio, further supporting the generation of S_v , which increased with prolonged H_2O_2 etching time. Quantitative analysis from ICP-OES and EPR (**Fig. R2c, d**) revealed an experimental value of S_v concentration is 1.54% for $Mn_{1-x}Zn_xS_v$, closely matching the optimal configuration screened by DFT.

Third, to further verify the consistency of the coordination environment of atomic Mn site, more precise fitting of the Mn K-edge EXAFS spectra was conducted (**Fig. R3**). The EXAFS fitting results show that the coordination number of Mn-S in the $Mn_{1-x}Zn_xS_v$ sample was stable at ~ 1.9 , and no Mn-Mn or Mn-O bond were observed, suggesting that S_v exists in isolation rather than in clusters.

Finally, following the reviewer's excellent suggestion, we performed spatial-resolved EELS mapping around individual Mn atoms. The oxidation state of Mn was determined by calculating the intensity ratio between the L3 (642.4 eV) and L2 (657.9 eV) peaks in the Mn L-edge EELS spectra (*Sci Rep*, **2023**, 13, 14132). The results confirm that Mn exists predominantly in the +2 oxidation state in $Mn_{1-x}Zn_xS_v$ (**Fig. R4**), consistent with XAS result, indirectly indicating that the existence state of S_v is isolated.

In the revised manuscript, we have included these results and added related discussion in the revised manuscript and the revised supplementary information (Fig. 3e, Supplementary Fig. 16 and 19, line 159-162, 173-174).

Fig. R1 Top-, side-view and ELF of the slab model of ZnS, Mn_1-ZnS and Mn_1-ZnS_v (111) surface (blue: Zn, yellow: S, red: Mn).

Fig. R2 (a) EPR spectra of the pristine Mn_1-ZnS and Mn_1-ZnS_v with different S-vacancy concentrations. (b) HAADF-STEM images, the corresponding EDS elemental mapping and atomic ratio (the number of M is the sum of Zn and Mn atoms) of Mn_1-ZnS_v with different

H₂O₂ etching time. (c) The linear fitting between ICP calculated S-vacancy and EPR quantified S-vacancy concentration over as-prepared catalysts. (d) ICP calculated M/S ratios as a function of the H₂O₂ etching treatment time (the number of M is the sum of Zn and Mn atoms).

Fig. R3 (a) Fourier-transform (FT) spectra of EXAFS oscillations for Mn foil, MnO, Mn₂O₃, Mn₁-ZnS and Mn₁-ZnS_v. (b) Mn K-edge EXAFS fitting analyses for Mn₁-ZnS_v samples in R Space.

Fig. R4 Mn-L_{2,3} electron energy loss spectrum (EELS) spectra for Mn₁-ZnS_v.

2. Only 0.6 wt % Mn is loaded; at this level, even a small fraction of MnO_x or MnS₂ clusters (<1 nm) could be missed by AC-STEM. A Mn K-edge EXAFS fitting range beyond 3 Å should be inspected for Mn–Mn contributions.

Response: Thank you for your constructive comments. In response to the reviewer's suggestion, the FT-EXAFS spectra of standard MnO and Mn₂O₃ was supplemented in **Fig. R3**, and the peak at 1.52 Å is attributed to Mn-O bond (*Angew. Chem. Int. Ed.* **2025**, 64, e202509741). No characteristic signals of the Mn-Mn and Mn-O bonds were detected, demonstrating that the Mn atoms are atomically dispersed in the Mn₁–ZnS_v, and no MnO_x forms are present.

To verify the contribution of the Mn-Mn bond, the second coordination shell of the Mn K-edge EXAFS over Mn₁–ZnS_v, beyond the range of 3 Å, was fitted in **Fig. R3** and **Table R1**. The EXAFS fitting results show that Mn atom mainly exists in the Mn-S path (the first coordination shell, 1.96 Å) and the relatively weak Zn-Mn path (the second coordination shell, 3.72 Å), without Mn-Mn path coordination. In addition, the independent Mn-Mn path was introduced for EXAFS fitting of Mn₁–ZnS_v, and the coordination number of Mn-Mn path was statistically less than 0.1, with no significant improvement in R-factor. It further confirms the atomic dispersion of Mn atoms and the absence of clusters of Mn compounds in Mn₁–ZnS_v (*Nat Commun* **2025**, 16, 4622).

In the revised manuscript, we have included these results in Fig. 3e and added related discussion in the line 173-174, 179-181 of the revised manuscript and Supplementary Fig. 19 of the revised supplementary information.

Table R1 EAXFS fitting results of the Mn K-edge for the Mn₁-ZnS_v.

Sample	Path	N	R/ Å	$\sigma^2/10^{-3}$ Å ²	$\Delta E/$ eV	R-factor
Mn ₁ -ZnS _v	Zn-S1	3.8±0.1	2.26±0.01	7.2±0.1	-1.9±0.1	0.009
	Mn-S	1.9±0.2	1.96±0.02	7.3±0.2	-1.5±1.1	
	Zn-Mn	11.8±0.2	3.72±0.01	6.8±0.2	-1.6±0.2	

3. Photocurrent and EIS (Supplementary Figs. 18–19) show improved charge-transfer kinetics for Mn₁-ZnS_v, but the incident-photon-to-current efficiency (IPCE) at 420 nm is not reported; this would directly corroborate the 8.1 % AQE claim.

Response: Thanks for your constructive comments. According to your suggestion, the incident photon-to-current efficiency (IPCE) was measured based on the photocurrent density under monochromatic light irradiation to directly evaluate the photoelectric conversion efficiency of Mn₁-ZnS_v. The IPCE can be calculated by using equation (1):

$$IPCE = \frac{\frac{hc}{\lambda} \times j}{Ie} \quad (1)$$

where j is the photocurrent density (A cm⁻²), I is the irradiation light intensity (W cm⁻²), λ is the equivalent wavelength (m), h is Planck's constant (6.626×10^{-34} J s⁻¹), c is the speed of light (2.998×10^8 m s⁻¹).

As shown in **Fig. R5a**, the corresponding IPCEs of pristine ZnS, Mn₁-ZnS and Mn₁-ZnS_v were calculated and consistent with the AQE results (**Fig. R5b**). Compared to the ZnS and Mn₁-ZnS, the Mn₁-ZnS_v showed an obvious enhancement of the IPCE in the range of 400–600 nm. At 420 nm, the IPCE of the Mn₁-ZnS_v exhibited a maximum value of 3.4%, which is 1.6 times higher than that of Mn₁-ZnS, implying excellent carrier transfer and separation. Then,

the AQE of 420 nm was calculated to be 8.12% in combination with the measurement parameters of IPCE. In conclusion, combining photocurrent, EIS, and IPCE, a solid and consistent verification of AQE results is provided from the kinetic perspective of charge separation and transport.

We have included the above results and added related discussions in the revised manuscript (the part of **Tests on photocatalytic CO₂ reduction with H₂O**) and the revised supplementary information (Supplementary Fig. 19).

Fig. R5 (a) IPCE plots of ZnS and Mn₁-ZnS, the Mn₁-ZnS_v. (b) The AQE of C₂H₄ evolution catalyzed by Mn₁-ZnS_v catalyst and the correlation between the related absorption spectra and absorption wavelengths.

4. Mott-Schottky plots give flat-band potentials, yet the absolute band positions (vs. NHE) are extracted assuming no Fermi-level pinning by Mn states; in situ ambient-pressure XPS under illumination could verify band alignment under operating conditions.

Response: We thank the reviewer for professionally reviewing our manuscript and giving constructive comments to improve our manuscript. Following the reviewer's insightful suggestion, to directly probe the band alignment in the working state, *in situ* ambient-pressure

VB-XPS experiments were performed under visible light irradiation. Under light irradiation, the valence band maxima for $\text{Mn}_1\text{-ZnS}$ and $\text{Mn}_1\text{-ZnS}_v$ were located at 1.00 and 0.55 eV below the Fermi level (**Fig. R6**), respectively, with a slight redshift compared to those in the dark, which is direct evidence for band bending caused by the accumulation of photogenerated carriers. Combining the results of UPS and UV-Vis, the conduction band minimum (CBM) of $\text{Mn}_1\text{-ZnS}$ and $\text{Mn}_1\text{-ZnS}_v$ were determined to be -1.16 and -1.23 eV, respectively. The CBM values of $\text{Mn}_1\text{-ZnS}$ (1.06 eV) and $\text{Mn}_1\text{-ZnS}_v$ (1.15 eV) obtained by Mott-Schottky analysis are very close to those obtained by *in situ* XPS measurement ($\Delta E < 0.1$ eV), indicating that the introduction of Mn did not cause severe Fermi-Level pinning.

In the revised manuscript, the electronic band structure versus Normal Hydrogen Electrode (NHE) was obtained by *in situ* XPS measurement (**Fig. R6**). The actual CBM under operating conditions is thermodynamically favorable for the $\text{CO}_2\text{-to-C}_2\text{H}_4$ conversion. We have included the above results in Supplementary Fig.s 22, 23 and added related discussions in the revised manuscript (line 198-202).

Fig. R6 *In situ* irradiated VB-XPS spectra of (a) $\text{Mn}_1\text{-ZnS}$ and (b) $\text{Mn}_1\text{-ZnS}_v$ in dark and irradiation.

Fig. R7 Schematic band diagram for ZnS, Mn₁-ZnS and Mn₁-ZnS_v catalysts.

5. D₂O experiments confirm water as the proton donor, but no KIE ($k_{\text{H}}/k_{\text{D}}$) for C₂H₄ formation is provided. A primary KIE would strengthen the claim that *CHO formation is turnover-limiting.

Response: We sincerely appreciate the reviewer's valuable comment and suggestion. In light of this suggestion, we have conducted the kinetic isotopic effect (KIE) of H/D over ZnS, Mn₁-ZnS and Mn₁-ZnS_v catalysts under the same reaction conditions (including light intensity, catalyst dosage, volume of reaction gas, etc.). Our DFT calculations (**Fig. R8a**) suggest that the *CO hydrogenation into *CHO may be the turnover-limiting step during the photocatalytic CO₂RR to C₂H₄. The activation of H₂O to active hydrogen species may thus be an important step that affects the *CHO formation. To understand the role of H₂O activation in photocatalytic CO₂RR to C₂H₄ product, we have measured the KIE of H/D over as-prepared catalysts. As displayed in **Fig. R8b**, the KIEs of H/D, which are defined as the ratio of C₂H₄ formation rates in H₂O and D₂O, were calculated to be 1.67, 1.36 and 1.18 over ZnS, Mn₁-ZnS and Mn₁-ZnS_v catalysts, respectively. It is generally believed that the KIE value of 2 is

manifested as a distinct primary KIE, indicating that the H₂O activation is involved in the rate-determining step (*Nat. Commun.* **2019**, 10, 892). By contrast, the KIE over the Mn₁-ZnS_v catalyst is close to 1. Therefore, the H₂O dissociation is not regarded as the rate-determining step over the Mn₁-ZnS_v catalyst. That is to say, the KIE further strengthens the claim that *CHO formation is turnover-limiting step. We have included the KIE of H/D in Supplementary Fig. 41 and added related discussion in the revised manuscript (line 228-269).

Fig. R8 (a) The free minimum-energy diagram for *CO hydrogenation and subsequent C-C coupling on Mn₁-ZnS_v (111) facet. (b) The C₂H₄ formation rate in H₂O and D₂O over pristine ZnS, Mn₁-ZnS and Mn₁-ZnS_v catalysts.

6. The H₂ evolution rate is suppressed to $\approx 1 \mu\text{mol g}^{-1} \text{h}^{-1}$, but its potential role as a competing pathway is not quantitatively modeled.

Response: Thank you sincerely for your constructive comments on our studies. The hydrogen evolution reaction (HER) is usually regarded as a competing reaction for the photoreduction reaction of carbon dioxide (CO₂RR). However, the adsorbed hydrogen species (*H) generated by H₂O dissociation, as a proton donor, is crucial for CO₂ photoreduction to hydrocarbons. To further explore the mechanism of the HER inhibition and the enhanced C₂H₄ selectivity, the competitive adsorption behavior of *H species and *CO intermediates on the catalyst was

investigated by simulation calculations. In our study, the introduction of low-coordinated Mn single atoms can lead to the nonequivalence of surface sites, thus modulating the distribution of *CO intermediates (Fig. R9). Our computational results show that the incorporation of low-coordinated Mn single atoms ingeniously balances the competitive adsorption of *CO intermediates and *H species (Fig. R10). Compared with *H species, *CO exhibits a stronger adsorption capacity (-1.36 eV) at the Mn active site in Mn₁-ZnS_v, which ensures that there are sufficient protons participating in the subsequent hydrogenation process without affecting the adsorption of key carbon-based intermediates at the active sites. Furthermore, the Gibbs free energy ($|\Delta G_{H^*}|$) for the HER on Mn₁-ZnS_v (0.56 eV) is higher than that of Mn₁-ZnS (0.22 eV) (Fig. R11), which not only effectively suppress the HER but also facilitate *CO protonation into *CHO. The kinetic barriers of *CO protonation (Fig. R12a) and the products yield (Fig. R12b) were conducted to validate that HER was not only suppressed but also ensured that the *H species was efficiently used for CO₂ reduction and its subsequent C-C coupling process. These results are supplement in the revised manuscript and supporting information (Supplementary Fig. 4, line 100-102, 245-248).

Fig. R9 The *CO adsorption energy at different sites of Mn₁-ZnS_v. The insets are top views of the structure of different sites and the distances between different Zn sites and Mn sites. The Mn, Zn, and S atoms are shown in pink, blue, and yellow colors, respectively.

Fig. R10 The comparison of *CO and *H adsorption energy of $\text{Mn}_1\text{-ZnS}$ and $\text{Mn}_1\text{-ZnS}_v$ with different sites (Mn: pink, Zn: blue, S: yellow).

Fig. R11 The Gibbs free energy diagrams for hydrogen adsorption of ZnS , $\text{Mn}_1\text{-ZnS}$ and $\text{Mn}_1\text{-ZnS}_v$.

Fig. R12 (a) Reaction energy diagram from *CO to *CHO on ZnS, Mn₁-ZnS and Mn₁-ZnS_v with molecular structures of initial, transition and final states inserted. (b) The products yield over ZnS, Mn₁-ZnS and Mn₁-ZnS_v catalysts.

7. Ten 4-h cycles (40 h total) show no loss, yet photocorrosion of ZnS in aqueous media is well-known. Longer-term tests (>100 h) or continuous-flow operation should be addressed.

Response: Thank you sincerely for pointing out our shortcomings. As shown in Fig. 5e, we have provided the photocatalytic stability tests for CO₂ photoreduction to C₂H₄ over 50 consecutive cycles (200 hours), during which no significant decline in activity or selectivity was observed. We apologize for the writing errors in the main text and have now corrected them (line 270).

8. Microwave/H₂O₂ etching is elegant for lab scale (~100 mg batches); scalability to gram or kilogram scale is not addressed.

Response: Thank you for your careful consideration. Scalability is the core issue that must be faced and solved from laboratory research to industrial application. In our study, microwave/H₂O₂ etching, as a novel method, was carried out to complete the proof of concept

and fundamental mechanism research on a lab scale (~100 mg). Small-scale experiments allow us to conduct rapid screening and optimization of a large number of parameters at a lower cost and with higher efficiency, which is crucial for understanding the nature of reactions and identifying key influencing factors.

In view of the scalability problem, a series of gram-level amplification experiments (1, 2 and 5 g) over Mn₁-ZnS in the microwave/H₂O₂ etching process were conducted under the condition of maintaining the same H₂O₂ concentration, precursor adding mode and microwave power. The products yield and C₂H₄ selectivity of the amplification experiments is illustrated in the **Fig. R13**. As the amount of Mn₁-ZnS was scaled up to gram level (1, 2 and 5 g), C₂H₄ yield and selectivity did not change significantly compared with 100 mg, and the C₂H₄ selectivity could still reach 99%. We will actively explore the reaction equipment and reaction parameters suitable for larger gram experiments, so as to provide a solid theoretical and experimental basis for the scale-up application of the catalyst. The corresponding discussion have been added in manuscript (Supplementary Fig. 34, line 237-239).

Fig. R13 The products yield and C₂H₄ selectivity over different amounts of Mn₁-ZnS (100 mg, 1 g, 2 g and 5 g) in microwave/H₂O₂ etching amplification experiments.

9. Only the ZnS(111) facet is modeled while ZnS also exposes (110) and (100) surfaces with different S_v formation energies. It will be necessary to model other two surfaces using DFT calculations for comparison.

Response: Thanks to the reviewer for highlighting this important issue and for your valuable suggestion. Based on your critical observation, we have accordingly performed DFT calculation of ZnS, Mn_1 -ZnS and Mn_1 -ZnS_v with exposed (110) and (100) facet, respectively. The structural models and vacancy formation energies of catalysts with different exposed facet were presented in **Fig. R14**. In contrast to Mn_1 -ZnS_v with exposed (111) facet, the S-vacancy (S_v) formation energies of Mn_1 -ZnS_v with exposed (110) and (100) facet were higher, indicating that S_v formed preferentially on the (111) facet (**Fig. R15**). In the revised manuscript, we have included these results in Supplementary Figs. 2, 3 and added the related discussions (line 90).

Fig. R14 Top-view and side-view of the slab model of ZnS, Mn₁-ZnS and Mn₁-ZnS_v with exposed (100) and (110) surface (blue: Zn, yellow: S, red: Mn).

Fig. R15 The vacancy formation energy (ΔE_{Vs}) of Mn₁-ZnS_v with (111), (100) and (110) facet, respectively.

10. Solvation and entropic contributions to *COCHO stability are not explicitly included.

Response: Thanks for your constructive suggestion. Ab-initio molecular dynamics (AIMD) simulations with fully explicit solvation were performed to elucidate solvation and entropic contributions to *COCHO stability. In our simulation model, we placed 202 water molecules, leading to the average density of $\sim 1 \text{ g}\cdot\text{cm}^{-3}$ to model the solvent water environment. The Gibbs free energy (ΔG) for the C-C coupling ($*\text{CO} + *\text{CHO} \rightarrow *\text{COCHO}$) on the Mn₁-ZnS_v surface reaction was calculated by

$$\Delta G = \Delta E + \Delta E_{ZPE} - T\Delta S - eU \quad (2)$$

in which ΔE_{ZPE} represent the change of the zero-point energy (ZPE), $T\Delta S$ is the entropic contributions (T was set to be 300 K), which were obtained from the vibrational frequency calculations through the VASPKIT code. e is the elementary charge and U is the conduction

band potential of semiconductor photocatalysts. The corresponding equilibrated structures obtained from the 5 ps AIMD simulation are presented in **Fig. R16a**.

Considering the solvation and entropy contributions, the simulation results further provide understanding on that a water environment close to the reaction condition can reduce the barrier of *CO – *CHO coupling into *COCHO from 0.62 eV to 0.54 eV (**Fig. R16b**) over Mn_1 – ZnS_v . This highlights the importance of considering the solvation environment and entropy contribution, especially for *COCHO stability, which forms strong hydrogen bonds with the interfacial water. Specifically, the Mn–C bonds between *COCHO and the interface shorten, leading to increased charge transfer between Mn and C and stronger stability of *COCHO (**Fig. R16c**). Collectively, taking into account the solvation and entropic effects, the energy barrier of C–C coupling decreased, further stabilizing the *COCHO intermediate and making the reaction pathway more thermodynamically and kinetically favorable.

The corresponding revision and discussion have been added in the revised manuscript (Supplementary Fig. 6, line 130-132, section on **Density functional theory (DFT) calculations**).

Fig. R16 (a) Side view of the simulation models of Mn_1 – ZnS_v . Hydrogen bonds are depicted by thin dashed lines. (b) Free energy diagram for C–C coupling ($^*CO + ^*CHO \rightarrow ^*COCHO$) on the Mn_1 – ZnS_v surface in vacuum and solvation environment, respectively (IS, TS, FS are

initial state, transition state, and final state, respectively). (c) The charge transfer between Mn and C atoms and the Mn–C bond length of $\text{Mn}_1\text{-ZnS}_v$ in vacuum and solvation environment. The zinc, manganese, sulfur, carbon, oxygen and hydrogen atoms are colored in blue, pink, yellow, gray, red and white, respectively.

11. ab-initio molecular dynamics simulations should be performed to capture dynamic vacancy reconstruction (e.g. 300 K).

Response: Thank you for the constructive and valuable comments. According to your excellent suggestion, ab-initio molecular dynamics (AIMD) simulations were performed for a duration of 10 ps at 300 K with time step of 1.0 fs (**Fig. R17**). As shown in **Fig. R17a**, time-resolved structural snapshots capture the vacancy structure remaining stable during the dynamic process without any collapse, diffusion or reorganization. The Mn-S bond length evolution (**Fig. R17b**) displays that in $\text{Mn}_1\text{-ZnS}_v$, the distance between the Mn active center and the S atom coordinated with it thermally oscillates only around the equilibrium value. The sulfur vacancies persisted throughout the AIMD simulations and no structural reconstruction was observed, which strongly proves that our proposed low-coordination Mn single atom structure is stable. These results are supplement in the revised manuscript and supplementary information (Supplementary Fig. 17, line 159-162, section on **Density functional theory (DFT) calculations**).

Fig. R17 (a) Representative snapshots from AIMD simulation for the vacancy reconstruction process. Blue, red and yellow balls represent Zn, Mn and S atoms, respectively; (b) Variations of Mn-S bond during the AIMD simulation.

12. Band-gap determination from UV-Vis (Fig. S13) uses a Tauc plot without stating whether direct or indirect transition is assumed; please clarify.

Response: Thank you very much for your rigorous consideration. The Tauc plot is derived based on the direct transition model, as ZnS is a well-known direct bandgap semiconductor. The bandgap energy (E_g) was determined by extrapolating the linear region of the plot of $(\alpha h\nu)^2$ versus photon energy ($h\nu$) (*Nano Energy* **2022**, 93, 106809). We have marked "direct transition" in the captions of Supplementary Fig. 20 in the supplementary information to make the analysis of UV-Vis more complete.

13. The reaction chamber volume, light intensity mapping, and temperature rise under focused Xe lamp are not documented; local heating can accelerate C–C coupling.

Response: Thank you for your careful check and suggestion. The photocatalytic reactions were carried out in Labsolar 6A device with a reactor volume of 370 mL and a system circulation pipeline volume of ~100 mL. In order to simulate the photosynthesis of natural plants, a 300 W Xe lamp was chosen as the light source. The average light intensity irradiated on the surface of the catalyst is $\sim 15 \text{ mW cm}^{-2}$ measured by a ThorLabs PM100D Power with a photodiode sensor.

In the photocatalytic experiment process, the temperature of the reaction system was continuously maintained at $25 \pm 1.5 \text{ }^\circ\text{C}$ through a cooling circulating water system. To address your concerns about "local heating", we conducted an additional verification experiment: a fine micro-thermometer was placed inside the reactor, adjacent to the catalyst area, to monitor the temperature changes under light irradiation in real time. As shown in the **Table R2**, under the same reaction light intensity, the measured local temperature change is extremely small ($\Delta T < 3.0 \text{ }^\circ\text{C}$). According to the Arrhenius equation, the influence of such a small temperature change on the formation rate of products can be ignored. The cooling circulating water system effectively removes the heat generated by light irradiation, maintaining the stability of the bulk phase temperature. In conclusion, through strict temperature control and real-time monitoring, it has been confirmed that the influence of local thermal effects can be ignored, and the observed activity originates from the photocatalytic process.

According to your suggestions, we have supplemented the above information in detail in the experimental section of the revised manuscript (section on **Tests on photocatalytic CO₂ reduction with H₂O**).

Table R2 The measured local temperature of catalyst surface during CO₂ reduction reaction.

Time (min)	Temperature (°C)
0	24.8
15	25.3
30	25.1
60	25.3
120	25.9
240	27.2

14. DRIFTS spectra are normalized to the ZnS phonon band at 1122 cm⁻¹, but absolute absorbance calibration would allow estimation of surface *CO coverage.

Response: Thank you for your meticulous suggestion. In our work, we normalized the *in situ* DRIFTS spectra to highlight the relative variation trends of the signal intensity of key intermediates (such as *CO, *COCHO, etc.) related to the reaction mechanism, thereby verifying the reaction path proposed. According to your suggestion, we have performed contour plot processing on the original *in situ* DRIFTS data of Mn₁-ZnS and Mn₁-ZnS_v with absolute absorbance values (**Fig. R18**), which is a key step to turn the mechanism demonstration from qualitative to quantitative. As shown in **Fig. R18**, the magnitude of the absolute absorbance signal at 2078 cm⁻¹, attributed to the *CO intermediate, can visually reflect the relative change of surface *CO coverage. Compared with Mn₁-ZnS, the *CO peak signal of Mn₁-ZnS_v is stronger during the photocatalytic CO₂RR. To further verify this result, CO-adsorbed DRIFTS was performed (**Fig. R19**), indicating a higher surface *CO coverage on Mn₁-ZnS_v. In addition, a correlation analysis was conducted between the changing trend of the absolute absorbance value of the *CO and the C₂H₄ formation rate and selectivity over Mn₁-ZnS and Mn₁-ZnS_v (**Fig. R20**). The favorable positive correlation trend presented indicates that a higher surface *CO coverage is conducive to improving the C-C coupling

efficiency, which is also the key reason for the high selectivity of C₂H₄. In response to the reviewer's suggestion, we have added a corresponding description to better evaluate surface *CO coverage and corroborate the key role of surface *CO coverage in driving C₂H₄ selectivity (Supplementary Fig. 30, line 221-225, 256-259).

Fig. R18 *In situ* DRIFTS for CO₂ photoreduction reaction with the absolute absorbance over Mn₁-ZnS (a) and Mn₁-ZnS_v (b).

Fig. R19 CO-adsorbed DRIFTS for Mn₁-ZnS (a) and Mn₁-ZnS_v (b).

Fig. R20 The correlation analysis between the changing trend of the absolute absorbance value of the *CO and the C₂H₄ formation rate and selectivity over Mn₁-ZnS and Mn₁-ZnS_v.

15. The error bars for AQE (currently single value at 420 nm) should be provided.

Response: We thank the reviewer for the good suggestion. As suggested, we have provided the error bars for AQE in Supplementary Fig. 39 of the revised supplementary information.

Fig. R21 The AQE of C₂H₄ evolution catalyzed by Mn₁-ZnS_v catalyst and the correlation between the related absorption spectra and absorption wavelengths.

Reviewer #2 (Remarks to the Author):

The manuscript addresses a significant challenge for the energy community – how to efficiently convert CO₂ into valuable C₂ products using solar energy directly. The manuscript is well-written, clearly organized, and contains a wide spectrum of characterization methods. My review will focus primarily on the photochemical characterization experiments, which require some clarifications and the filling of missing data points to assess the paper's merit accurately.

Response: We sincerely thank your positive feedback and for recognizing the significance of our work. We are also grateful for the constructive comments and valuable suggestions, which have helped us enhance the quality of the manuscript. Below, we have provided detailed responses to your comments. The point-to-point response is as follows and related changes are marked in **RED** in the revised manuscript and the supplementary information.

1. The paper's main point is that using low-coordination Mn sites to stabilize CO₂RR intermediates results in a higher efficiency towards C₂₊ products, as otherwise these critical intermediates are not able to react further. To support this claim, the authors should also perform control experiments using CO as feed for the ZnS catalyst and add this to Fig 5c. This is a critical point to address.

Response: Thanks to the reviewer for highlighting this important issue and for your valuable suggestion. Following the suggestion, we performed control experiments by supplying 100% CO as the reactant under otherwise identical reaction conditions over the ZnS catalyst.

As shown in **Fig. R22**, the ZnS catalyst exhibits markedly lower C₂H₄ yield with CO as the reactant compared to CO₂, while also underperforming significantly against the Mn₁-ZnS catalyst under the same conditions. This result underscores that the low-coordination Mn sites not only stabilize the CO intermediate, but also facilitate CO₂ activation, thereby forming a

highly active adsorbed *CO species favorable for subsequent C-C coupling. The synergistic "activation-stabilization" mechanism is the essential reason for the high selectivity of the Mn₁-ZnS_v catalyst in generating C₂H₄ product under CO₂ reduction reaction.

In response to the reviewer's suggestion, we have added the related explanations in the revised manuscript (Fig. 5c, line 256-259).

Fig. R22 The products yield over ZnS, Mn₁-ZnS and Mn₁-ZnS_v catalysts using CO₂ and CO as reactant, respectively.

2. Secondly, the authors claim that they excelled in two metrics important for photochemical systems: ethylene production rate and ethylene selectivity. Given this is a primary claim, the authors should provide more details on how the experiments were conducted and how the metrics were calculated. Also, were the experiments repeated for reproducibility? The error bars should be added to the photochem experiments. The authors should also include their calculations of apparent quantum efficiency (specific values used, not only the formula).

Response: We sincerely appreciate the valuable comments. The specific details on experimental part and performance indicators have been provided point by point.

(1) Based on your suggestions, we have supplemented and explained more details regarding (i): **Photochemical testing**: The photocatalysts (0.2 g) was dispersed at the bottom of watch glass (diameter is ~4.0 cm) with 5 mL deionized water. The mixed sample was uniformly ultrasound and put into the photoreaction system with a reactor volume of 370 mL. Before introduction of light irradiation, the photoreaction system was thoroughly vacuum-treated (3 times), and then pure CO₂ was injected into the internal circulation reactor for 30 min. And a cooling circulating water device was utilized to maintain the reaction system a constant temperature at 25 ± 0.5 °C. In order to simulate the photosynthesis of natural plants, one xenon lamp (300 W) was chosen as the light source, and added a 380 nm external filter to make the light source only retain the visible light. The light intensity was measured by a ThorLabs PM100D Power with a photodiode sensor (~15 mW cm⁻²). After starting the light source and the gas chromatograph automatic sampling device, the sample was analyzed every 15 min, and the reaction test was carried out for 4 h. (ii) **Yield and selectivity calculations**: The real-time peak area of the products was recorded during the photocatalytic test, and the product yield was obtained by the standard working curve. The selectivity of CO₂ reduction to carbon products (S_{CO_2}) is obtained by the following formula (3):

$$S_{CO_2}(\%) = \frac{12n(C_2H_4)+2n(CO)+8n(CH_4)+ 2n(CHOOH)}{12n(C_2H_4)+2n(CO)+8n(CH_4)+ 2n(CHOOH)+2n(H_2)} \times 100\% \quad (3)$$

The selectivity of C₂H₄ reduction is obtained by the following formula (4):

$$S_{C_2H_4}(\%) = \frac{12n(C_2H_4)}{12n(C_2H_4)+2n(CO)+8n(CH_4)+ 2n(CHOOH)} \times 100\% \quad (4)$$

(2) The repeated photoactivity test was carried out and an error bar for C₂H₄ yield was added in **Fig. R23**.

Fig. R23 The evolution of C₂H₄ product during photocatalytic CO₂ reduction over time.

(3) The ratio of the number of photons converted by the reaction to the number of incident photons is the value of AQE, and the specific formula is as follows:

$$AQE_{C_2H_4} = \frac{12 \times \text{Number of reacted electrons}}{\text{Number of incident photons}} \times 100 \% = \frac{12 \times n_{C_2H_4} \times N_A}{S \times I \times t \times \frac{\lambda}{h \times c}} \times 100 \% =$$

$$\frac{12 \times 76.6 \times 0.2 \times 4 \times 10^{-6} \times 6.022 \times 10^{23}}{\pi \times 2^2 \times 15 \times 10^{-3} \times 14400 \times \frac{420 \times 10^{-9}}{6.626 \times 10^{-34} \times 2.998 \times 10^8}} = 0.0812 = 8.12\% \quad (5)$$

where S is the irradiated area (cm²), I is the irradiation light intensity (W cm⁻²), t is the irradiation time (s), λ is the equivalent wavelength (m), h is Planck's constant (6.626 × 10⁻³⁴ J s⁻¹), c is the speed of light (2.998 × 10⁸ m s⁻¹) and N_A is Avogadro's number (6.022 × 10²³ J mol⁻¹). n_{C₂H₄} is the amount of C₂H₄ produced in 4 h.

In the revised manuscript, we have included these results in Fig. 5b and added the related discussions in the method section (Supplementary Table 4 and section on **Tests on photocatalytic CO₂ reduction with H₂O**).

3. Important details are missing in the experimental description -e.g., what was the total amount of CO₂ injected into the reactor, and how was the photocatalyst immobilized?

Response: We thank the reviewer for this valuable suggestion. The photocatalytic reactions were carried out in a reactor volume of 370 mL and a system circulation pipeline volume of ~100 mL. The pure CO₂ was injected into the internal circulation reactor (50 mL min⁻¹) until the system pressure reached 80 kPa. The photocatalyst power (0.2 g) was dispersed in deionized water (5 mL) and sonicated to form a homogeneous slurry at the bottom of watch glass (diameter is ~4.0 cm), and place this glass in the reactor.

We have supplemented this information in the revised manuscript to enhance its rigor and repeatability (section on **Tests on photocatalytic CO₂ reduction with H₂O**).

4. Fig 5d: control condition 5, should it be “replacing CO₂ with Ar” instead of “replacing Ar with CO₂”?

Response: Thanks for pointing out this mistake. We have revised “replacing CO₂ with Ar” in the revised manuscript (Fig. 5d).

5. What do authors mean as “efficiency of tourism” (SI, section on Tests on photocatalytic CO₂ reduction with H₂O)? there is also a typo in Eq1. Looks like this was an auto-correct mistake, please review all your writing carefully.

Response: Thank the reviewer for checking our work carefully and providing valuable comments. We sincerely apologize for the obvious clerical error in "efficiency of tourism" here, which should be "turnover frequency". For better expression, we have made corrections in the corresponding section: “The ratio of the number of photons converted by the reaction to the number of incident photons is the value of AQE”. We have conducted a comprehensive review of all formulas, charts and main text to ensure that such errors have been completely corrected.

Reviewer #3 (Remarks to the Author):

Response: Thanks for your professional comments and constructive suggestions. Those comments are all valuable and very helpful to us. We have tried our best to improve the manuscript and make the corresponding modification in the revised manuscript. These revisions do not affect the framework of this paper. We sincerely hope that these modifications and explanations will be acknowledged. Once again, thank you very much for your comments and suggestions.

Reviewer #4 (Remarks to the Author):

Tang et al. report a Mn single-atom catalyst embedded in ZnS with sulfur vacancies ($\text{Mn}_1\text{-ZnS}_v$) for photocatalytic CO_2 conversion to C_2H_4 . Photocatalysts were modified by inclusion of Mn, hypothesizing that “single-atom” sites with Mn would increase binding energy for $^*\text{CO}$ and steer selectivity to C_2 products, Figure 1. Nanoparticle catalysts were made by solution-based method: ZnS and Mn-ZnS with 3 different mass ratios. Fe, Co, Ni, and Cu were also employed as dopants. S vacancies were induced by microwave treatment for 5-15 seconds: Zn- ZnS_v series. While a large number of samples appear to have been made, study focuses on one of them (page 6): “Inductively coupled plasma optical emission spectrometry (ICP-OES) confirmed a Mn content of 0.6 wt% (Supplementary Table 1).” It does appear, EPR in Figure S8, that this sample has more unpaired electrons than control which did not undergo the microwave treatment. A very high selectivity for this preparations is claimed: $76.6 \mu\text{mol}\cdot\text{g}^{-1}\cdot\text{h}^{-1}$ under sacrificial-agent-free conditions.

No sacrificial species were employed and ^{13}C and ^2D labeling was used to ensure that carbon and protons come from CO_2 and water, respectively. O_2 production is also reported, Figure 22. Study thus follows “best practices” for experimental reports of this type, see 10.1021/acscenergylett.8b00196 and 10.1002/anie.201207199. Also, interestingly, $\text{Mn}_1\text{-ZnS}_v$ is able to convert CO to C_2H_4 (this is not always the case for CO_2RR photocatalysts). They estimate an AQE of 8% at 420 nm. The authors attribute the performance to $^*\text{CO}$ stabilization and $^*\text{CO}\text{-}^*\text{CHO}$ coupling, supported by in situ spectroscopy and DFT calculations.

Review prompts are addressed as follows:

What are the noteworthy results?

The performance of the photocatalysts, both in terms of activity/selectivity, is very good compared to other reports in the literature, Figure 5f.

Will the work be of significance to the field and related fields? How does it compare to the established literature? If the work is not original, please provide relevant references.

While the performance metrics are high, the novelty is limited. Similar catalyst architectures and comparable activity/selectivity have been reported (e.g., *Energy Environ. Sci.* 17, 5060–5069 (2024); *J. Am. Chem. Soc.* 145, 422–435 (2023)), and the present study does not clearly distinguish itself conceptually or mechanistically.

Does the work support the conclusions and claims, or is additional evidence needed?

In part, see detailed comments below.

Are there any flaws in the data analysis, interpretation and conclusions? Do these prohibit publication or require revision?

Revisions are needed in the data analysis, see detailed comments below.

Is the methodology sound? Does the work meet the expected standards in your field?

Yes. Again, the use of isotope labeling, CO substitution, etc. are best practices in the field.

Is there enough detail provided in the methods for the work to be reproduced?

Yes.

Summary statement.

The manuscript reports strong catalytic performance but lacks sufficient novelty and mechanistic depth to meet the standards of Nature Communications. Addressing the above points, particularly clarifying the Mn–vacancy interplay, rigorously validating the universality

claim, and strengthening stability and mechanistic evidence, would be essential to enhance the originality and impact of the work.

Response: We would like to extend our deepest gratitude to the reviewer for your detailed and thorough feedback. We have taken all the concerns seriously and tried our best to address them by numerous additional experiments and/or substantial revisions. As you will see in the revised manuscript and the detailed point-by-point responses, your concerns about the consistency of experimental data with theoretical calculations should have been fully addressed. In the current version, the scientificity of this work have been significantly improved compared to the previous submission. Please see our point-by-point responses as follows.

Detailed comments.

1. The interplay between sulfur vacancies and Mn single atoms is insufficiently resolved. Figure S26 presents 4 h C₂H₄ yields for Fe, Co, Ni, and Cu-doped catalysts subjected to the microwave treatment. Yields are ~70-120 $\mu\text{mol/g}$. What is not shown on this graph is the Mn-ZnS not subjected to the microwave treatment: 190 $\mu\text{mol/g}$. This data does not support the statement: “indicating a universal enhancement in photocatalytic performance induced by low-coordination metal centers.” So it is not clear whether Mn substitution, which appears to be the original hypothesis, or the S vacancies is the main effect. At the very least, a ZnS_v control (without Mn) should be synthesized, characterized, tested, and included in DFT calculations. Figures 4a and 4b do not convincingly isolate the effect of sulfur vacancies. Full catalytic cycle calculations with corresponding energy profiles are needed for both cases, and any synergistic effect should be explicitly demonstrated in the mechanism.

Response: We thank the reviewer for the insightful suggestion. As suggested, we have conducted corresponding supplementary experiments and calculations to clarify the interaction

between the Mn single atom and sulfur vacancies. The following are our point-by-point responses:

(1) ZnS_v sample was synthesized according to exactly the same microwave treatment method (without the addition of Mn precursors) and characterized, tested and DFT calculated (**Fig. R24**). The clear signal at $g=2.005$ and the relative red shift of XPS indicate that sulfur vacancies have been successfully created in ZnS_v . The absorption edge of ZnS_v is redshifted relative to pure ZnS, further confirming the successful construction of vacancies. Photocatalytic CO_2 reduction performance reveals that ZnS with sulfur vacancies (ZnS_v) exhibits negligible enhancement in C_2H_4 yield and selectivity (7.3%) compared to pristine ZnS (5.6%) (**Fig. R25**). In stark contrast, Mn single atom-doped ZnS ($\text{Mn}_1\text{-ZnS}$) achieves a substantial improvement in C_2H_4 selectivity (74.5%), demonstrating that isolated sulfur vacancies are inactive for C_2H_4 generation. Instead, the introduction of doped metal atoms, establishing bimetallic sites, is essential for activating the C_2H_4 pathway. Remarkably, $\text{Mn}_1\text{-ZnS}_v$ synthesized *via* vacancy engineering from $\text{Mn}_1\text{-ZnS}$ precursor delivers near-unity C_2H_4 selectivity (99.1%), which unambiguously identifies low-coordination Mn single atoms as the pivotal active centers driving high C_2H_4 selectivity.

(2) To further explain the general applicability of low-coordination transition metal single-atom doping in ZnS materials, we supplemented the photocatalytic performance of saturated-coordination $\text{M}_1\text{-ZnS}$ ($\text{M} = \text{Fe}, \text{Co}, \text{Ni}, \text{Cu}$) and low-coordination $\text{M}_1\text{-ZnS}_v$ in **Fig. R26**. All $\text{M}_1\text{-ZnS}_v$ variants exhibited substantially higher C_2H_4 yields than their saturated-coordination $\text{M}_1\text{-ZnS}$ counterparts, indicating a certain enhancement in photocatalytic performance induced by similar low-coordination metal centers. The results of general applicability test further emphasize that the introduction of another metal site is a necessary condition for generating C_2H_4 , and a low-coordination environment is a key strategy driving the improvement of C_2H_4 selectivity.

(3) Based on your suggestions, the free energy of the entire reaction path from CO_2 to C_2H_4 were carried out over ZnS , ZnS_v , $\text{Mn}_1\text{-ZnS}$ and $\text{Mn}_1\text{-ZnS}_v$. As shown in the **Fig. R24f**, the energy barrier of the ZnS_v path is not significantly lower than that of ZnS and the main product is still CO . Both $\text{Mn}_1\text{-ZnS}$ and $\text{Mn}_1\text{-ZnS}_v$ enhance the adsorption of $^*\text{CO}$ intermediates, indicating that the Mn active sites facilitate the reduction of hydrogenation energy barriers. Crucially, the low-coordination $\text{Mn}_1\text{-ZnS}_v$ exhibits the lowest energy barriers for $^*\text{CO}$ protonation ($^*\text{CHO}$ formation) and the asymmetric coupling of $^*\text{CO}$ - $^*\text{CHO}$ ($^*\text{COCHO}$ formation). It demonstrates that S_v acts as “electronic modulators”, synergizing with Mn centers to optimize the C-C coupling pathway, thereby enabling high C_2H_4 yield and selectivity.

In the revised manuscript, we have corrected and supplemented the relevant discusses to elaborate on the synergistic effect between the Mn active center and sulfur vacancies (Supplementary Figs. 35, 36 and 46, line 241-242, 245-248, 295-298).

Fig. R24 (a) Top-, (b) side- view of the slab model of ZnS_v (blue: Zn, yellow: S, red: Mn). (c) EPR spectra of ZnS_v , $\text{Mn}_1\text{-ZnS}$ and $\text{Mn}_1\text{-ZnS}_v$. XPS spectra of (d) Zn 2p and (e) UV-Vis DRS over ZnS and ZnS_v . (f) Reaction pathways and C-C coupling step for CO_2 photoreduction over ZnS , ZnS_v , $\text{Mn}_1\text{-ZnS}$ and $\text{Mn}_1\text{-ZnS}_v$.

Fig. R25 The photocatalytic CO₂ reduction performance over pristine ZnS, ZnS_v, Mn₁-ZnS and Mn₁-ZnS_v catalysts.

Fig. R26 The products yield of CO₂ reduction based photocatalytic reaction with different transition metal elements over M₁-ZnS and M₁-ZnS_v (M=Mn, Fe, Co, Ni, Cu) catalysts under 4 h irradiation.

2. The assertion that multiple metals (Fe, Co, Ni, Cu) can be incorporated as single sites requires direct evidence. At least one alternative metal (e.g., Cu) should be systematically compared with Mn via in situ DRIFTS, XAS, and full DFT energy profiles, etc., with a deeper discussion of performance differences beyond the d-band center argument.

Response: Thank you to the reviewer for the valuable comment. As suggested, we have systematically selected Cu as the representative contrast metal, and completed the following in-depth supplementary research.

To probe the local coordination environment of Cu atoms in $\text{Cu}_1\text{-ZnS}_v$, X-ray absorption spectroscopy (XAS) was conducted. X-ray absorption near-edge structure (XANES) spectra (**Fig. R27a**) showed that the absorption edge position of $\text{Cu}_1\text{-ZnS}_v$ lies between those of the Cu_2O and CuO reference standards, indicating that Cu exists in $\text{Cu}^{\delta+}$ ($1 < \delta < 2$). The Cu *K*-edge extended X-ray absorption fine structure (EXAFS) spectra in *R*-space (**Fig. R27b**) reveal an absence of the Cu-Cu bond peak at 2.2 Å, while displaying a prominent peak at 1.8 Å, corresponding to the Cu-S bond in the first coordination shell, which strongly supports the formation of Cu SAs. Further EXAFS fitting analysis determined the Cu-S coordination number to be 2.0, and the Cu-S bond length to be 2.13 Å, which is longer than that of Mn-S bond (1.96 Å). The above characterization collectively indicates the successful synthesis of $\text{Cu}_1\text{-ZnS}_v$ structure.

To elucidate the underlying mechanism of performance differences between the $\text{Cu}_1\text{-ZnS}_v$ and $\text{Mn}_1\text{-ZnS}_v$, *in situ* DRIFTS spectra, which are sensitive to the real-time reaction intermediates, were measured. As illustrated in **Fig. R28**, a distinct vibrational signal was detected at 2078 cm^{-1} , corresponding to linearly adsorbed CO ($^*\text{CO}_L$). In contrast, the $\text{Cu}_1\text{-ZnS}_v$ exhibited significantly weaker $^*\text{CO}_L$ absorption peaks under identical conditions, suggesting that $\text{Mn}_1\text{-ZnS}_v$ featured a superior ability to adsorb $^*\text{CO}$ intermediates for further C-C coupling. This strengthened adsorption was attributed to the stronger $d \rightarrow 2\pi$ backbonding between the d-orbital of low-coordination Mn atoms in the $\text{Mn}_1\text{-ZnS}_v$ and the $2\pi^*$ antibonding orbital of $^*\text{CO}$, which is more pronounced than for low-coordination Cu atoms in the $\text{Cu}_1\text{-ZnS}_v$. The stronger affinity of $\text{Mn}_1\text{-ZnS}_v$ for $^*\text{CO}$ facilitated its further protonation into $^*\text{CHO}$ and coupling into $^*\text{COCHO}$, as manifested by the peak at 1315 cm^{-1} , which served as a critical

intermediate in the pathway to C_2H_4 formation. The performance differences, supported by theoretical calculations (**Fig. R29**), could be attributed to the higher ΔG associated with the *CO -to- *CHO conversion for Cu_1-ZnS_v (0.82 eV) than for Mn_1-ZnS_v (0.62 eV), indicating comparatively less favorable energetics of Cu_1-ZnS_v . In addition, the free energy for *CO - *CHO coupling on Mn_1-ZnS_v is significantly lower than that of on Cu_1-ZnS_v , which suggests that the stronger free energy driving force on Mn_1-ZnS_v likely contributes to its enhanced catalytic activity.

In the revised manuscript, we have included these results in Supplementary Figs. 45, 49, 50 and added the related discussions (line 293-298 and supplementary information).

Fig. R27 (a) Normalized Cu K-edge XANES spectra and (b) the corresponding Fourier spectra in R -space of Cu_1-ZnS_v , Cu_2O , CuO and Cu foil. (c) FT-EXAFS fitting curves of Cu_1-ZnS_v .

Fig. R28 *In situ* DRIFTS for CO_2 photoreduction reaction over Cu_1-ZnS_v (a) and Mn_1-ZnS_v (b) catalysts.

Fig. R29 Free energy diagrams of the CO₂ reduction pathway to C₂H₄ on Cu₁-ZnS_v and Mn₁-ZnS_v, respectively.

3. Figure 2. In the free energy diagram, conversion of CO₂(g) to *CO is shown to have a negative free energy (in principle, spontaneous). What conditions were assumed to make this reaction downhill. I suspect that that the authors are using the computational hydrogen electrode that the reaction is proceeding via proton-coupled electron transfers. If this is the case, what potential was assumed and why was it though that this could be realized in photocatalytic experiments?

Response: We appreciate the reviewer's suggestion to clarify the CO₂(g)-to-*CO reaction processes. Photocatalytic CO₂ reduction reaction (CO₂RR) is a complex proton-coupled electron transfers (PCET) process, and its photocatalytic efficiency is influenced by various factors such as thermodynamics and kinetics. The computational hydrogen electrode (CHE) model was employed to obtain the free energy of the elementary step in the theoretical research on CO₂RR (*J. Phys. Chem. B*, **2004**, 108, 17886–17892.). The whole elementary steps of photocatalytic CO₂-to-*CO conversion could be described as following:

In the photocatalytic process, the generation of electron-hole pairs by Mn₁-ZnS_v catalyst under light irradiation is a prerequisite for the occurrence of photocatalytic reactions. The potential at the conduction band minimum of the Mn₁-ZnS_v catalyst is sufficiently negative than the equilibrium potential of the reaction to drive the CO₂ reduction reaction. In the actual reaction, although the electrons come from the photogenerated electrons in the excited state of the Mn₁-ZnS_v catalyst and the H⁺ originates from the cleavage of water molecules in solution, the energy of the proton–electron pair is equal to half the energy of the hydrogen molecule according to the computational hydrogen electrode method (*J. Electrochem. Soc.* **2005**, 152, J23). Thus, the Gibbs free energy of the reaction can be obtained by the following equation:

$$\Delta G = \Delta E + \Delta E_{\text{ZPE}} - T\Delta S - eU \quad (9)$$

in which ΔE_{ZPE} represent the change of the zero-point energy (ZPE), $T\Delta S$ is the entropic contribution (T was set to be 300 K), which were obtained from the vibrational frequency calculations through the VASPKIT code. e is the elementary charge and U is the conduction band potential of semiconductor photocatalysts. Specifically, the calculation formulas are as follows:

$$\Delta G[*\text{CO}_2] = G[*\text{CO}_2] - G(*) - G(\text{CO}_2) \quad (10)$$

$$\Delta G[*\text{COOH}] = G[*\text{COOH}] - G(*) - G(\text{CO}_2) - 0.5 \times G(\text{H}_2) \quad (11)$$

$$\Delta G[*\text{CO}] = G[*\text{CO}] + G(\text{H}_2\text{O}) - G(*) - G(\text{CO}_2) - G(\text{H}_2) \quad (12)$$

Based on the above discussion, photocatalytic CO₂-to-*CO conversion on the Mn₁-ZnS_v surface with an appropriate *CO adsorption energy is thermodynamically favorable. The corresponding supplementary discussion have been added in the revised manuscript (section on **Density functional theory (DFT) calculations**).

4. More generally, the influence of light on the catalytic process, particularly from a mechanistic perspective, is not discussed in sufficient detail. DFT calculations, especially those related to Supplementary Fig. 3, should incorporate the effect of photoexcitation to better reflect the actual reaction conditions.

Response: We sincerely thank the reviewer for the constructive comments to further elucidate the photocatalytic mechanism.

Firstly, to explore the light absorption capacity, the band structure of the prepared $\text{Mn}_1\text{-ZnS}_v$ photocatalyst, as determined by UV-Vis DRS, VB-XPS, UPS and Mott-Schottky plots (Supplementary Figs. 20-25), possesses a suitable conduction band potential (-1.23 eV) for $\text{CO}_2\text{-to-C}_2\text{H}_4$ and valence band potential (1.65 eV) for $\text{H}_2\text{O-to-O}_2$, enabling the generation of electron-hole pairs. This is the essential first step that provides the thermodynamic driving force (-eU in our DFT model) for the CO_2 reduction reaction.

Upon irradiation, photogenerated electrons are excited to the conduction band. EIS and transient photocurrent measurements (Supplementary Fig. 28) revealed that low-coordination Mn single atom site exhibited significantly enhanced charge separation and migration efficiency compared to saturated-coordination Mn single atom site. What's more, TRPL decay curves and SPV (Supplementary Fig. 27) were carried out to confirm the long lifetime of carriers and a high concentration of photogenerated electrons accumulated on the $\text{Mn}_1\text{-ZnS}_v$ catalyst surface.

To gain insight into the electron transfer mechanism under light illumination, the *Quasi-in situ* XPS measurement of $\text{Mn}_1\text{-ZnS}_v$ was studied (Fig. R30). Under illumination, *Quasi-in situ* XPS analysis reveals a distinct positive shift for S 2p accompanied by a negative binding energy shift for Mn 2p, providing direct spectroscopic evidence that photogenerated electrons

are transferred from S (electron donor) to Mn (electron trap). Subsequently, photoelectrons are rapidly injected into the adsorbed CO₂ molecules (**Fig. R31**) through the electron transport channel. Activated CO₂ undergoes a series of proton-coupled electron transfer (PCET) processes to obtain *CO intermediate (**Fig. 2b**). The localized electron-rich environment of low-coordinated Mn single-atom site enhances the adsorption of *CO and drives the multi-electron pathway toward selective C₂H₄ formation (**Supplementary Fig. 7**).

Finally, in the photocatalytic process, the generation of electron-hole pairs by Mn₁-ZnS_v catalyst under light irradiation is a prerequisite for the occurrence of photocatalytic reactions. The potential at the conduction band minimum of the Mn₁-ZnS_v catalyst is sufficiently negative than the equilibrium potential of the reaction to drive the CO₂ reduction reaction. Considering the effect of photoexcitation on free energy to better reflect the actual reaction conditions, -eU represents the thermodynamic driving force for Mn₁-ZnS_v to utilize photogenerated electrons (*Nat Catal*, **2025**, 8, 728–739), and the energy of the proton–electron pair is equal to half the energy of the hydrogen molecule according to the computational hydrogen electrode method (*J. Electrochem. Soc.* **2005**, 152, J23). Thus, the Gibbs free energy of the reaction can be obtained by the following equation:

$$\Delta G = \Delta E + \Delta E_{\text{ZPE}} - T\Delta S - eU \quad (13)$$

in which ΔE_{ZPE} represent the change of the zero-point energy (ZPE), $T\Delta S$ is the entropic contribution (T was set to be 300 K), which were obtained from the vibrational frequency calculations through the VASPKIT code. e is the elementary charge and U is the conduction band potential of semiconductor photocatalysts.

In response to the reviewer’s suggestion, we have added the related explanations in the revised manuscript (Supplementary Fig. 26, line 198-208, section on **Density functional**

theory (DFT) calculations).

Fig. R30 *Quasi-in situ* XPS measurement of (a) Mn 2p and (b) S 2p_{3/2} over Mn₁-ZnS_v in dark and in light.

Fig. R31 CO₂-TPD profiles of ZnS, Mn₁-ZnS and Mn₁-ZnS_v.

5. Supplementary Fig. 25 is ambiguous. Given the claimed 200-hour stability (Fig. 5e), post-reaction characterizations, especially XAS, should be shown for the spent catalyst. More frequent GC sampling (e.g., every 5–10 min) is recommended to capture potential fluctuations.

Response: We thank the reviewer for the valuable suggestion. The validation of catalyst stability and experimental details are essential to enhance the rigor and persuasive power of

this work. The following is a point-by-point response to your comment.

(1) The structural integrity of $\text{Mn}_1\text{-ZnS}_v$ over several cycles was evaluated and the results were shown in Supplementary Fig. 43. SEM images (Supplementary Fig. 43a) and XRD (Supplementary Fig. 43b) show that the crystallite morphology and crystal structure remain intact after extended operation, respectively. The consistent XPS spectra of fresh (Supplementary Fig. 43c-f) and reused (Supplementary Fig. 43g, h) samples further support the chemical stability.

(2) To confirm that local coordination environment of Mn atoms in spent sample, XAS was conducted. There are no characterized signals of Mn-Mn bond in $\text{Mn}_1\text{-ZnS}_v$ after reactions and the coordination number of Mn atoms remains at 1.9 ± 0.3 after long-term reaction (**Fig. R32**). The stability of the coordination environment ensures the efficient photocatalytic performance of the $\text{Mn}_1\text{-ZnS}_v$ catalyst.

(3) As you suggest, we have conducted more frequent GC sampling (every 15 min) to evaluate the photocatalytic activity and selectivity of the $\text{Mn}_1\text{-ZnS}_v$ catalyst. As shown in **Fig. R33**, the photocatalytic stability of $\text{Mn}_1\text{-ZnS}_v$ was evaluated over 50 consecutive cycles (200 h), during which no significant decline in activity or selectivity was observed.

Following your suggestion, the corresponding revision and discussion have been added in the revised manuscript (Supplementary Figs. 42-44, line 274-279).

Fig. R32 (a) Normalized Mn K-edge XANES spectra and (b) Fourier-transform (FT) spectra of EXAFS oscillations of Mn₁-ZnS_v, the spent Mn₁-ZnS_v and the related reference samples.

Fig. R33 The cycling tests for CO₂ photoreduction to C₂H₄ over Mn₁-ZnS_v photocatalyst under the same as reaction conditions.

6. ¹³CO labeling, combined with in situ DRIFTS, would provide stronger evidence for the proposed asymmetric *CO-*CHO coupling pathway.

Response: Thank you for the insightful comments. As suggested, we have conducted the *in situ* DRIFTS tests with labeled ¹³CO as adsorbate to further verify the asymmetric *CO-*CHO

coupling over $\text{Mn}_1\text{-ZnS}_v$ catalyst during photocatalytic reduction reaction. *In situ* DRIFTS of $\text{Mn}_1\text{-ZnS}_v$ exhibit characteristic isotope effect peaks involving $^*\text{CO}$ and $^*\text{CHO}$ species (Fig. R34). Furthermore, the increasing tendency of the characteristic isotope effect peaks related to $^*\text{COCHO}$ intermediates, $^{*13}\text{CO}^{13}\text{CHO}$, was observed over $\text{Mn}_1\text{-ZnS}_v$ with the light irradiation during the photocatalytic process, which strongly supports the asymmetric coupling of $^*\text{CO}$ and $^*\text{CHO}$. Combined with the results of GC-MS of photocatalytic $^{13}\text{CO}_2$ reduction with D_2O over $\text{Mn}_1\text{-ZnS}_v$ catalyst, the formation of $^{13}\text{C}_2\text{D}_4$ further provides strong evidence for the asymmetric coupling of $^*\text{CO}$ - $^*\text{CHO}$.

In response to the reviewer's suggestion, we have added the related explanations in the revised manuscript (Supplementary Fig. 31, line 221-225).

Fig. R34 *In situ* DRIFTS of ^{13}CO chemisorption over $\text{Mn}_1\text{-ZnS}_v$ catalyst during photocatalytic reduction reaction.

7. Figure S24. This is said to be a GC-MS spectrum. But multiple analytes are present. Were chromatograms summed to produce this data?

Response: Thank you to the reviewers for raising the important question to give us the opportunity to clarify our data analysis methods more clearly. Supplementary Fig. S24 shows

the conversion of profile spectrum into centroid spectrum in the gas chromatography-mass spectrometry (GC-MS) coupled analysis. The GC-MS spectra reflect the relationship between the m/z of all ion fragments and their relative abundance rather than later artificial summary of chromatograms, and is used to visually infer the structure of the target products of photocatalytic $^{13}\text{CO}_2$ reduction with D_2O over $\text{Mn}_1\text{-ZnS}_v$ catalyst. The presence of multiple analytes in the mass spectrum corresponds to characteristic ion fragments with different m/z, and the product was identified as C_2H_4 by comparison with the NIST standard mass spectrometric library. We have supplemented relevant content in the method section of the revised manuscript to eliminate misunderstandings (line 264-265, section on **Characterization**).

8. The following is said about Figure 3b. “Elemental mapping via energy-dispersive X-ray spectroscopy (EDS, Fig. 3b) showed homogeneous distribution of Zn, Mn, and S throughout the sample, with Mn atoms clearly penetrating into the ZnS matrix.” On what basis can it be said that the Mn distribution is homogenous? What would an inhomogeneous distribution look like? Also “Mn atoms clearly penetrating into the ZnS matrix.” It is not possible to see if Mn atoms are inside the ZnS with TEM as it produces a 2D projection of a 3D object.

Response: We thank the reviewer for the critical observations regarding the interpretation of our EDS results, which help us present our findings with greater scientific rigor.

We fully agree that a qualitative assessment from a single EDS elemental mapping is insufficient to definitively claim “homogeneous distribution”. In response to your concern, line-scan profiles were carried out to offer further insights into the spatial distribution of Zn, Mn, and S throughout the sample (**Fig. R35**). EDS elemental mapping combined with line-scan profiles provides strong evidence for the uniform distribution of elements (*Nat. Mater.* **2024**, 23, 1355–1362; *ACS Catal.* **2024**, 14, 17, 13400–13407). Meanwhile, we have

modified from "homogeneous distribution" to a more accurate description: "Elemental mapping via energy-dispersive X-ray spectroscopy (EDS, **Fig. 3b**) and the STEM-EDS line scanning profiles (Supplementary Fig. 12) corroborated the even distribution of Zn, Mn, and S throughout the sample, with no observable aggregation of Mn into large clusters or particles."

The conventional EDS provides a 2D projection of a 3D object and cannot directly resolve the internal location of dopant atoms. To confirm the integration of Mn and ZnS lattices, the results of XRD, XPS and EXAFS were further discussed. The absence of any diffraction peaks corresponding to Mn oxide or sulfide phases, coupled with the slight red shift in ZnS peak positions (**Fig. R36a**), indicates lattice incorporation. Then, the binding energy of Mn 2p peaks confirm that Mn exists in the oxidation state of +2 (**Fig. R36b**), which is consistent with the expectation of replacing the Zn^{2+} site. In addition, EXAFS fitting revealed that the first-shell Mn–S coordination number was 3.7 for $\text{Mn}_1\text{-ZnS}$ (**Fig. R36c**), which is one of the direct pieces of evidence proving that Mn atoms have replaced Zn atoms and exists within the ZnS lattice.

In response to the reviewer's suggestion, we have corrected the original manuscript and added the related explanations in the revised manuscript (Supplementary Fig. 12, line 148-151).

Fig. R35 STEM image of Mn_1-ZnS (a) and Mn_1-ZnS_v (b), and the corresponding STEM-EDS elemental line scan (c, d), respectively.

Fig. R36 (a) XRD pattern of pristine ZnS, Mn_1-ZnS and Mn_1-ZnS_v . (b) XPS spectra of Mn 2p over Mn_1-ZnS and Mn_1-ZnS_v . (c) EXAFS spectrum fitting result of Mn_1-ZnS in R space.

9. Figure 3c caption. What do the yellow circles mean and what quantitative criterion was used to draw them?

Response: We thank the reviewer for the comment. Aberration-corrected annular dark-field scanning transmission electron microscope (ADF-STEM) image was employed to further confirm the atomic dispersion of Mn in $\text{Mn}_1\text{-ZnS}_v$ catalyst. As shown in **Fig. R37**, several isolated dark spots (marked by yellow circles) can be captured by utilizing the Z-contrast difference between the lighter Mn and the heavier Zn atoms, which indicate successful atomic substitution of Zn by Mn atoms. A line-scan intensity profile across a representative region (Profile 1–2, **Fig. R37**) reveals localized intensity dips, confirming atomic-level dispersion of Mn atoms (*Angew. Chem. Int. Ed.* **2025**, e202516903).

We have included the above results in Fig. 3c and added the related discussions in the line 150-154 in the revised manuscript.

Fig. R37 HAADF-STEM image of $\text{Mn}_1\text{-ZnS}_v$ with Mn single atoms marked by yellow circles, and atomic intensity corresponding to the marked area of the line profile 1 and profile 2.

10. Page 6. “with directional sulfur vacancies (S_v).” What is a directional sulfur vacancy? ZnS has the zinc-blende structure, Figure 6. There is only one type of S in this structure and just one type of S vacancy. Also, how can a vacancy have a direction (vectorial?).

Response: Thank you to the reviewer for raising the important issue, which provides us with an opportunity to clarify a key innovation point in our work. From a crystallographic perspective, there is indeed only one type of S vacancy. The expression "directional" in our original manuscript might not be precise enough to cause a misunderstanding. We have corrected "directional sulfur vacancies" to "targeted sulfur vacancies". Specifically, S vacancies are preferentially created in specific chemical environments (around Mn single-atom sites) via innovative selective-etching strategy.

The innovative selective etching strategy refers to the selective breaking of chemical bonds. Microwave thermally activated hydrogen peroxide (H_2O_2) generates hydroxyl radicals ($\cdot OH$) radicals that preferentially attack and break the Mn-S bonds, with weaker bond energy, to create targeted vacancies at single-atom coordination sites (*J. Am. Chem. Soc.* **2025**, 147, 3, 2689–2698). Therefore, the regulation of the spatial distribution of vacancies is achieved through thermodynamically and kinetically controlled synthesis, enriching vacancies around the designed active sites, thereby constructing the expected low-coordination Mn active sites.

We have made corresponding revisions to the manuscript (Supplementary Fig. 9, line 138-141).

11. Page 7. “...revealed a S_v concentration of 1.54% for Mn_1-ZnS_v , closely matching the theoretical value predicted by DFT.” How can DFT predict the vacancy concentration produced by the microwave treatment?

Response: Thank the reviewer for highlighting this valuable issue, allowing us to clearly clarify the innovative design concept of combining theoretical guidance and experimental results in our work. We fully agree that traditional DFT calculations indeed cannot directly predict the precise defect concentration produced by microwave treatment. The word "predict" in the manuscript may be a misleading expression. We sincerely apologize for this and have made corrections in the revised manuscript.

In fact, our research approach embodies a strategy of " theory-guided directional design and experimental verification ", which is specifically described as follows:

(1) **The role of theoretical calculation is to screen the optimal configuration rather than predict the synthesis concentration.** We systematically constructed Mn single-atom doped ZnS models with different coordination structures and evaluated their C-C coupling capabilities by key descriptors (such as *CO adsorption energy). The DFT results revealed that a specific low-coordination (N=2) Mn active site, with a sulfur vacancy (S_v) concentration of 1.33%, theoretically exhibits the optimal adsorption strength for *CO intermediates, which is the most favorable for C-C coupling. The concentration value of 1.33% is a "target optimal value", which stems from the theoretical exploration of the structure-activity relationship.

(2) **The goal of experimental synthesis is directed toward the optimal configuration.** Based on the above theoretical screening results, we developed and optimized the microwave-induced H_2O_2 etching, a targeted defect engineering strategy, aiming to prepare the $Mn_{1-x}Zn_xS_v$ catalyst with near-ideal vacancy concentrations.

(3) **The validation of experimental results demonstrated the exceptional alignment between theory, structure, and performance.** The experimental S_v concentration in the $Mn_{1-x}Zn_xS_v$ catalyst prepared was determined to be 1.54% by multiple characterization methods (ICP,

EPR, EDS Mapping), which was highly consistent with the theoretical target value (1.33%). Moreover, the Mn₁-ZnS_v catalyst demonstrates outstanding C₂H₄ yield and selectivity.

Therefore, the optimal structure model was screened out through theoretical calculation, and then the catalyst with key parameters highly consistent with the ideal model was successfully prepared *via* precise synthesis experiments. The closed-loop from theoretical design to experimental implementation is precisely one of the core innovative points of our work, which significantly surpasses the traditional catalyst development model.

We have corrected the relevant expressions and added related discussions in the revised manuscript (line 138-141, 159-162, section on **Density functional theory (DFT) calculations**).

12. Page 9. “In contrast, Mn₁-ZnS_v exhibited a stronger *CO signal at 2078 cm⁻¹ (Fig. 4b), indicating enhanced generation and stabilization of *CO.” Authors cannot conclude that CO₂ -> CO is faster on the Mn-doped material. The steady-state concentration of *CO depends both on the generation and desorption rate.

Response: We sincerely thank the reviewer for this insightful comment, which correctly highlights the distinction between the steady-state concentration of an intermediate and its rate of formation. We agree that a stronger *CO signal in *in-situ* DRIFTS could indeed result from either a faster formation rate or a slower desorption rate. The corresponding expression is modified as below: “In contrast, Mn₁-ZnS_v exhibited a stronger *CO signal at 2078 cm⁻¹ (**Fig. 4b**), indicating enhanced stabilization and coverage of *CO.”

To further provide strong evidence for the enhanced stabilization of *CO, performance experiments and theoretical calculations were carried out. DFT calculations (**Fig. R38a**) demonstrate that the *CO adsorption energy is stronger on the Mn₁-ZnS_v compared to other structures, which is consistent with the high steady-state of *CO. In addition, Compared with

Mn₁-ZnS, the significantly increased C₂H₄ yield and decreased CO yield on Mn₁-ZnS_v (Fig. R38b) indicate that more *CO intermediates are involved in the C-C coupling pathway, which further explains the enhanced stabilization of *CO observed in *in-situ* DRIFTS experiment. In response to the reviewer's suggestion, we have revised the related explanations in line 221-225 in revised manuscript.

Fig. R38 (a) Adsorption energies of *CO on Mn₁-ZnS and Mn₁-ZnS_v. (b) Product yields of products (HCOOH, CO, C₂H₄ and CH₄) and the selectivity of C₂H₄ product over ZnS, Mn₁-ZnS and Mn₁-ZnS_v catalysts.

Reviewer #5 (Remarks to the Author):

The paper reports on the Mn₁-ZnS_v photocatalyst for CO₂ reduction to C₂H₆. The study employs theoretical guidance in experimental design, and the overall structure is complete, with clear logic and theoretical foundations. However, the following points require further discussion:

Response: We sincerely appreciate the reviewer's detailed summary of our work, and are pleased that the novelty and strengths of our designed Mn₁-ZnS_v photocatalyst and its near-unity selectivity for CO₂-to-C₂H₄ photoconversion were well recognized. We have carefully addressed each of the reviewer's concerns below.

1. On page 5, the statement "The low-coordination Mn site in Mn₁-ZnS_v strengthens the Mn-CO π backbonding" – what does this mean? Please provide a detailed explanation. What is the relationship between this and the differential charge density analysis?

Response: Thanks to the reviewer for highlighting this important issue and for your valuable suggestion. We are sorry for our unclear or inadequate descriptions. Typically, π backbonding is a pivotal concept in metal-ligand bonding (**Fig. R39**). In the Mn-*CO interaction, it involves two synergistic electron processes: σ -donation and π -backbonding (π -backdonation). (1) The CO molecule donates its lone pair electrons from the carbon atom to fill an empty d-orbital of the Mn atom, forming a σ -bond. (2) The Mn atom, in turn, backdonates electrons from its filled d-orbitals into the π -antibonding orbital of CO (*ACS Catal.* 2024, 14, 9, 7011–7019). In our manuscript, low-coordination Mn sites, characterized by higher electron density and an electron-rich state, significantly enhance the strength of the π -backbonding process.

The enhanced π backbonding was directly verified through Bader charge and charge density difference analyses (**Fig. R40**). Compared to Mn₁-ZnS, more electron transfer of Mn₁-ZnS_v

clearly demonstrates that low-coordination Mn sites strengthen π -backbonding. Furthermore, the intensified π backbonding significantly enhances the adsorption strength of the $^*\text{CO}$ intermediate on the low-coordination Mn sites, while simultaneously activating the adsorbed $^*\text{CO}$ species to a higher degree for subsequent C–C coupling.

We have supplemented the corresponding descriptions and the changes are highlighted in red in the revised manuscript (line 97-100, 119-122).

Fig. R39 Illustration of the interaction between the 5σ and $2\pi^*$ of $^*\text{CO}$ and metal surfaces.

Fig. R40 Differential charge density and Bader charge of $^*\text{CO}$ adsorbed on $\text{Mn}_1\text{-ZnS}_v$ (a) and $\text{Mn}_1\text{-ZnS}$ (b) (the isosurface is $0.005 \text{ e}/\text{\AA}^3$). Blue and purple represent charge gaining and losing, respectively.

2. To ensure greater rigor, in addition to calculating the adsorption of *CO on the Mn atom in both Mn₁-ZnS and Mn₁-ZnS_v structures, the authors should also consider the adsorption on Zn atoms surrounding the Mn site. Although *CO does not readily adsorb on Zn atoms in pure ZnS, the electronic structure modulation induced by Mn may alter the adsorption behavior on neighboring Zn atoms.

Response: Thanks to the reviewer for this important and meaningful suggestion. We agree that in single-atom catalyst systems, the electronic modulation effect of the central metal atom on its surrounding coordination environment is crucial and may significantly alter the adsorption behavior of adjacent sites. The adsorption energies of the *CO intermediate at different adsorption sites on the Mn₁-ZnS_v model have been systematically compared. **Fig. R41** shows that the adsorption energy of *CO at the Mn site is much lower than that at the surrounding Zn sites, indicating that the modulation of Mn is insufficient to cause strong adsorption of *CO at the Zn sites. It was further verified that the low-coordination Mn site stabilizes the *CO intermediate through an enhanced Mn-*CO π -backbonding, creating the necessary conditions for the subsequent C-C coupling. In the revised manuscript, we have included the above results in Supplementary Fig. 4 and added the related discussions in line 97-102 in the revised manuscript.

Fig. R41 The *CO adsorption energy at different sites of Mn₁-ZnS_v. The insets are top views of the structure of different sites and the distances between different Zn sites and Mn sites. The Mn, Zn, and S atoms are shown in pink, blue, and yellow colors, respectively.

3. On page 6, why can microwave radiation directionally induce defects? What advantages does microwave irradiation offer over other synthesis methods specifically for ZnS materials? How does H₂O₂ etch S? Is it possible that O atoms are introduced into ZnS during this process?

Response: Thank you to the reviewer for raising these key questions related to the synthetic mechanism. We have supplemented these key points in detail, and the specific explanations are as follows:

(1) Innovative selective-etching strategy were adopted to synthesize Mn₁-ZnS_v catalyst via a microwave irradiation-H₂O₂ etching treatment under an Ar atmosphere. Microwave thermally activated hydrogen peroxide (H₂O₂) generates hydroxyl radicals (\cdot OH) radicals that selectively cleave metal-S bonds, with weaker bond energy, to create targeted vacancies at single-atom coordination sites (*J. Am. Chem. Soc.* **2025**, 147, 3, 2689–2698). The microwave energy is uniformly and transiently injected into the bulk phase under the Ar atmosphere, resulting in a uniform defect distribution during the H₂O₂ etching process.

(2) In contrast to conventional external heating methods, microwave energy directly interacts with the molecular dipoles of ethylene glycol to achieve rapid, bulk phase heating of the material. Specifically, i) High Efficiency and Energy Saving: The reaction rate is significantly accelerated compared to conventional hydrothermal/solvothermal methods, substantially reducing synthesis time; ii) Uniform Size Distribution: The bulk phase heating characteristic facilitates the formation of more homogeneous size distributions, avoiding the issue of particle size heterogeneity caused by thermal gradients in traditional approaches. iii) Morphology and

Size Control: The extremely rapid nucleation kinetics generally yield smaller nanocrystals, thereby exposing larger active surface areas.

(3) The essence of H₂O₂ etching is an oxidation process, that is, $\text{ZnS} + \text{H}_2\text{O}_2 \rightarrow \text{Zn}^{2+} + \text{S}^{2-} + 2\text{OH}^-$. Subsequently, S²⁻ can be further oxidized to soluble species (such as SO₃²⁻, SO₄²⁻), thereby removing S atoms from the lattice to obtain sulfur vacancies (S_v). Theoretically, it is possible for a small amount of O²⁻ or OH⁻ in solution to be incorporated into the lattice during the etching process.

To address this concern regarding the possible introduction of O atoms into the ZnS lattice during the H₂O₂ etching process, we have performed the following additional characterizations and provided a detailed discussion in the revised Supplementary Information: i) XPS analysis: As shown in **Fig. R42**, the O 1s XPS spectra were deconvoluted into two primary peaks at 531.5 and 532.4 eV, corresponding to the surface adsorbed oxygen (O_{ads}) and oxygen in adsorbed water (O_w) (*Proc. Natl. Acad. Sci.* **2024**, 121 (11) e2319427121), respectively. Crucially, the lattice oxygen in metal oxide at 530.0 eV wasn't observed, indicating that the doping of O atoms can be disregarded. ii) XRD analysis: We re-examined the XRD patterns of Mn₁-ZnS before and after H₂O₂ treatment (**Fig. R43**). No new diffraction peaks corresponding to any zinc oxide (ZnO) or zinc oxysulfide (ZnO_xS_y) phases were detected. Based on the results, H₂O₂ acts as an oxidant and etching agent for generating sulfur vacancies, rather than a source for O-doping.

In response to the reviewer's suggestion, we have added a corresponding description in Supplementary Fig. 9 of the supplementary information.

Fig. R42 XPS spectra of O 1s of Mn₁-ZnS and Mn₁-ZnS_v catalysts.

Fig. R43 XRD pattern of pristine ZnS, Mn₁-ZnS and Mn₁-ZnS_v.

4. In Figure 3e, how can it be confirmed that the bond is Mn-S rather than Mn-O?

Response: Thank you for your critical comment to confirm the precise chemical structure of the Mn active center. We supplemented the Fourier transform spectra of EXAFS of standard MnO and Mn₂O₃, and compared them with Mn₁-ZnS and Mn₁-ZnS_v catalysts. As shown in Fig. R44, MnO and Mn₂O₃ both exhibit the characteristic peak of the Mn-O bond at 1.52 Å. In contrast, a prominent peak at 1.96 Å was assigned to Mn-S bonding in Mn₁-ZnS and Mn₁-

ZnS_v catalysts. The significant peak position difference is the most direct and powerful evidence for distinguishing Mn-O bonds from Mn-S bonds. The absence of Mn-Mn and Mn-O coordination in both Mn₁-ZnS and Mn₁-ZnS_v indicates that Mn atoms are atomically dispersed and coordinated with neighboring S atoms.

In the revised manuscript, we have included these results in Fig. 3e and added related discussion in the line 173-174 of the revised manuscript.

Fig. R44 Fourier-transform (FT) spectra of EXAFS oscillations for Mn foil, MnO, Mn₂O₃, Mn₁-ZnS and Mn₁-ZnS_v.

5. In Figure 3c, given that the atomic masses of Mn and Zn are very similar and the Mn₁-ZnS_v is bulk rather than ultrathin sheet-like, it is generally challenging to distinguish them via ADF-STEM. Can the authors provide a reasonable explanation for this observation?

Response: Thank the reviewer very much for checking our manuscript and making professional comments!

Aberration-corrected annular dark-field scanning transmission electron microscope (ADF-STEM) image was employed to further confirm the atomic dispersion of Mn in $\text{Mn}_1\text{-ZnS}_v$ catalyst. As shown in **Fig. R45**, several isolated dark spots (marked by yellow cycles) can be captured by utilizing the Z-contrast difference between the lighter Mn and the heavier Zn atoms, which indicate successful atomic substitution of Zn by Mn atoms. A line-scan intensity profile across a representative region (Profile 1–2, **Fig. R45**) reveals localized intensity dips, confirming atomic-level dispersion of Mn atoms (*Angew. Chem. Int. Ed.* **2025**, e202516903). In addition, EDS elemental mapping images and STEM-EDS line scanning profiles show a homogeneous dispersion of Zn, Mn, and S atoms throughout the $\text{Mn}_1\text{-ZnS}_v$ sample (**Fig. R46**).

More importantly, EXAFS fitting revealed that the existence of first-shell Mn–S path and the absence of Mn–Mn path and Mn–O path in $\text{Mn}_1\text{-ZnS}_v$ catalyst (**Fig. R47**), which provided strong evidence for the ADF-STEM observation.

According to the reviewer's comments, the detailed discusses have been added in the revised manuscript. (Figs. 3c and 3e, Supplementary Fig. 12, line 148-154).

Fig. R45 HAADF-STEM image of $\text{Mn}_1\text{-ZnS}_v$ with Mn single atoms marked by yellow circles, and atomic intensity corresponding to the marked area of the line profile 1 and profile 2.

Fig. R46 (a) EDS elemental mapping of $\text{Mn}_1\text{-ZnS}_v$. (b) STEM image of $\text{Mn}_1\text{-ZnS}_v$ and (c) the corresponding STEM-EDS elemental line scan.

Fig. R47 Fourier-transform (FT) spectra of EXAFS oscillations for $\text{Mn}_1\text{-ZnS}$ and $\text{Mn}_1\text{-ZnS}_v$.

6. Please include the energy band structures related to other possible CO_2 reduction products to further support the claim that CO_2 is more readily reduced to C_2H_4 .

Response: We thank the reviewer for this suggestion regarding the energy band structures to deepen the understanding of C_2H_4 product selectivity. We have included the energy band structures related to possible CO_2 reduction products in **Table R3**. The energy band position of the photocatalyst is crucial for determining the thermodynamic feasibility of the CO_2RR . In this system, the relatively negative conduction band position of $\text{Mn}_1\text{-ZnS}_v$ (-1.23 eV) indicates its strong reducing ability, which is sufficient to trigger the reduction products such as C_2H_4 currently studied by the researchers. However, the separate energy band structure is not enough to explain the dynamics of specific C_2H_4 selectivity. To provide a more rigorous and mechanistic explanation for the high C_2H_4 selectivity, we have performed and included additional theoretical calculations of the Gibbs free energy for the possible products from adsorbed *CO , including HCHO , CH_3OH , CH_4 , and C_2H_4 . The calculations reveal that the C-C coupling step ($\text{*CO} + \text{*CHO} \rightarrow \text{*CHOCO}$) on the low-coordination Mn site in the $\text{Mn}_1\text{-ZnS}_v$ model exhibits a significantly lower energy barrier compared to the other potential

reaction pathways (**Fig. R48**). In conclusion, the energy band position determines the thermodynamic feasibility of CO₂-to-C₂H₄ conversion over Mn₁-ZnS_v, while the unique local coordination environment of the low-coordination Mn site promotes C-C coupling kinetically. In response to the reviewer's suggestion, we have added the related explanations in line 198-202 and Supplementary Fig. 24 of the supplementary information.

Table R3 Photocatalytic reduction of CO₂ and its standard redox potential.

Reaction process	Potential/ E^\ominus (V) vs. NHE (298 K)
$\text{CO}_2 + 2\text{H}^+ + 2\text{e}^- \rightarrow \text{CO} + \text{H}_2\text{O}$	-0.53
$\text{CO}_2 + 8\text{H}^+ + 8\text{e}^- \rightarrow \text{CH}_4 + 2\text{H}_2\text{O}$	-0.24
$\text{CO}_2 + 12\text{H}^+ + 12\text{e}^- \rightarrow \text{C}_2\text{H}_4 + 4\text{H}_2\text{O}$	-0.38
$\text{CO}_2 + 6\text{H}^+ + 6\text{e}^- \rightarrow \text{CH}_3\text{OH} + \text{H}_2\text{O}$	-0.39
$\text{CO}_2 + 4\text{H}^+ + 4\text{e}^- \rightarrow \text{HCHO} + \text{H}_2\text{O}$	-0.48
$\text{CO}_2 + 2\text{H}^+ + 2\text{e}^- \rightarrow \text{HCOOH}$	-0.61

Fig. R48 The free minimum-energy diagram for *CO hydrogenation and subsequent C-C coupling on Mn₁-ZnS_v (111) facet.

7. Since water is present in the photocatalytic reaction system, were any products detected in the aqueous solution?

Response: We sincerely thank the reviewer for your concern. We have conducted the ^1H nuclear magnetic resonance (NMR) analysis for the liquid product. For pristine ZnS, formic acid (HCOOH) ($12.1 \mu\text{mol g}^{-1} \text{h}^{-1}$) was detected, and its selectivity was 8.9%. No liquid products were detected on the $\text{Mn}_1\text{-ZnS}$ and $\text{Mn}_1\text{-ZnS}_v$ catalysts above the instrument's detection limit. These experimental results provided further insight into the high C_2H_4 selectivity of the $\text{Mn}_1\text{-ZnS}_v$ catalyst. We have added the related discusses in the revised manuscript (Fig. 5a, line 253).

8. It would be beneficial if the authors could calculate the consumption and balance of photogenerated carriers throughout the reaction to further elucidate the stability of the catalyst.

Response: We appreciate the reviewer's suggestion to investigate the consumption and balance of photogenerated carriers throughout the reaction to gain a thorough understanding of the photochemical stability of $\text{Mn}_1\text{-ZnS}_v$ catalyst. In our system, the main half-reaction is as follows:

To quantify the consumption and equilibrium of photogenerated carriers, that is, the total electron consumption in the reduction reaction and the hole consumption in the H_2O oxidation reaction, the number of electrons transferred in the formation of the two half-reaction products were precisely measured. The ratio of the number of electrons transferred in the oxidation reaction to the number of electrons transferred in the reduction reaction is defined as $R_{\text{O/R}}$. The closer the $R_{\text{O/R}}$ value is to 1, the better the photochemical stability of the catalysts. As shown in **Table R4**, the $R_{\text{O/R}}$ values of ZnS, $\text{Mn}_1\text{-ZnS}$ and $\text{Mn}_1\text{-ZnS}_v$ are all close to 1, which means that the consumption of photogenerated electrons and holes on the catalysts' surface is nearly

balanced, directly avoiding the oxidation corrosion or photocorrosion of the catalysts caused by charge accumulation. Thus, it explains the reason for the excellent photochemical stability of the catalysts from the perspective of mechanism. In addition, the cyclic stability test (Fig. 5e) and the structural characterization before and after the reaction (Supplementary Figs. 43 and 44) in the manuscript also confirmed the internal reasons for the excellent photostability of Mn₁-ZnS_v. In response to the reviewer's suggestion, we have added the related discusses in the revised manuscript (Supplementary Table 4, line 276-279).

Table R4 The formation rates and the value of R_{O/R} over ZnS, Mn₁-ZnS and Mn₁-ZnS_v catalysts.

Sample	Formation rate [$\mu\text{mol g}^{-1} \text{h}^{-1}$]						R _{O/R} [*]
	Reduction					Oxidation	
	C ₂ H ₄	CO	CH ₄	HCOOH	H ₂	O ₂	
ZnS	1.3	64.6	12.9	12.1	113.5	130.7	1.05
Mn ₁ -ZnS	47.5	28.4	17.3	N.d. ^d	39.2	238.3	1.13
Mn ₁ -ZnS _v	76.6	4.2	N.d.	N.d.	5.2	255.2	1.09

$$* R_{O/R} = \frac{4 \times n_{O_2}}{12 \times n_{C_2H_4} + 2 \times n_{CO} + 8 \times n_{CH_4} + 2 \times n_{HCOOH} + 2 \times n_{H_2}}$$

Response to Reviewers' Comments

Manuscript number: NCOMMS-25-62587A

Title: Near-unity CO₂-to-ethylene photoconversion over low-coordination manganese single-atom sites on binary sulfide

A Point-to-point Response to Reviewer's comments

Dear Editor and Reviewers,

Thank you for checking our manuscript and proposing valuable comments! The comments are highly appreciated and helpful to improve our work. We have made corresponding changes to the manuscript and Supplementary information (**highlighted in yellow in the revised versions**) to address the reviewers' concerns; these changes are specified and discussed in a point-to-point response to the reviewers' comments, as shown below:

REVIEWER COMMENTS

Reviewer #1 (Remarks to the Author):

Following a comprehensive re-evaluation of the manuscript, it has been concluded that the method for free energy calculation remains flawed. Specifically, the use of the electrochemical potential (U) in equation 5 is inappropriate, as it represents the conduction band potential of the semiconductor photocatalyst, which is not applicable in this context. Furthermore, in equation 3, the energy should correspond to the entire catalyst surface slab rather than just the surface. Additionally, the manuscript does not provide sufficient details regarding the calculation of the free energy barrier. In light of these significant issues, I regret that I cannot

recommend this work for publication in its current form.

Response: We sincerely appreciate the reviewer's insightful and constructive comments. Your suggestions accurately identified the key issues in the free-energy calculation methodology and its expression in our original manuscript, and they have provided valuable guidance for improving the scientific rigor of this work. In response, we have added relevant theoretical references, clarified the calculation framework, and corrected all free-energy results to ensure that every concern is fully resolved. We respectfully hope that the revised version may be given a new opportunity for consideration in *Nature Communications*.

1. Addressing concerns about the “-eU” item in the equation 5 for free energy

(i) We fully agree with the reviewer that the use of the electrochemical potential (U) in equation 5 was inappropriate under the computational context of this work. After careful examination, we recognize that your judgment is scientifically sound and directly points to the direction needed for methodological revision.

In our preliminary theoretical screening, we evaluated two parallel computational schemes: (1) the classical Gibbs free-energy formalism, $\Delta G = \Delta E + \Delta E_{\text{ZPE}} - T\Delta S$ (Nat Commun 2024, 15, 10589, *J. Am. Chem. Soc.* **2023**, 145, 51, 28276, *Sci. Adv.* **2020**, 6, eaaz2060), and (2) a modified form incorporating $-eU$, $\Delta G = \Delta E + \Delta E_{\text{ZPE}} - T\Delta S - eU$ (*Nat Catal* **2025**, 8, 728, *Nat Catal* **2023**, 6, 574, *J. Chem. Phys.* **2013**, 139, 044103), where U represents the conduction-band potential of the semiconductor and $-eU$ approximates the thermodynamic driving force from photogenerated electrons.

Based on your valuable comments, we re-examined the applicability of the two calculation schemes. Considering the research system and computational scenarios, the introduction of “-eU” in the equation 5 for free energy indeed cannot represent a complete photodynamic description process with inherent complexity. It only offers a balanced and computationally tractable approach for initial mechanistic insight and identifying activity trends. The generally

recognized classical Gibbs free energy equation ($\Delta G = \Delta E + \Delta E_{ZPE} - T\Delta S$) is more in line with the free energy calculation logic of this study. Therefore, we have corrected all the calculation results involving Gibbs free energy in the original manuscript to those using the classical Gibbs free energy equation (Fig. R1).

To further verify the reliability of the revised conclusion, we compared the Gibbs free energy diagram calculated by the two equations, and found that the trends of the theoretical screening results of the two were completely consistent. The core theoretical guiding conclusion was not affected, ensuring the robustness of the research conclusion.

Fig. R1 a, The free energy diagram for CO₂ reduction reaction to display the conversion of CO₂-to-*CO over ZnS, Mn₁-ZnS and Mn₁-ZnS_v. **b**, The free minimum-energy diagram for *CO hydrogenation and subsequent C-C coupling on Mn₁-ZnS_v (111) facet. **c**, Reaction energy diagram from *CO to *CHO on ZnS, Mn₁-ZnS and Mn₁-ZnS_v with molecular structures of initial, transition and final states inserted. **d**, Reaction pathways and C-C coupling step for CO₂ photoreduction over ZnS, Mn₁-ZnS and Mn₁-ZnS_v.

(ii) The considerations for using equation 5 in the original manuscript are as follows. In the photocatalytic process, the generation of electron-hole pairs by $\text{Mn}_1\text{-ZnS}_v$ catalyst under light irradiation is a prerequisite for the occurrence of photocatalytic reactions. The potential at the conduction band minimum of the $\text{Mn}_1\text{-ZnS}_v$ catalyst is sufficiently negative than the equilibrium potential of the reaction to drive the CO_2 reduction reaction. The methodology presented in equation 5 is based on the Computational Hydrogen Electrode (CHE) model, a standard method in electrocatalysis (*J. Phys. Chem. B* **2004**, 108, 17886–17892). It explicitly introduces the electrode potential (U) into the Gibbs free energy calculation to describe the reaction free energy at a given potential. To account for the influence of photoexcitation on reaction free energy and better approximate realistic conditions, the conventional computational hydrogen electrode (CHE) model was extended to photocatalytic system. In our work, the "electrode potential (U)" is redefined as the effective conduction band potential of the semiconductor photocatalysts, while $-eU$ represents the thermodynamic driving force provided by photogenerated electrons in $\text{Mn}_1\text{-ZnS}_v$ (*J. Chem. Phys.* **2013**, 139, 044103). This adaptation allows for a self-consistent mapping of the thermodynamic landscape in photocatalytic systems. Consequently, the $-eU$ correction is introduced to the standard expression, yielding a photocatalytic free energy equation: $\Delta G = \Delta E + \Delta E_{\text{ZPE}} - T\Delta S - eU$, with the core rationale supported by references, such as *Nat Catal* **2025**, 8, 728–739, *Nat Catal* **2023**, 6, 574–584, *J. Chem. Phys.* **2013**, 139, 044103.

(iii) According to your valuable suggestions and a large number of literatures on photocatalytic CO_2 reduction (*Nat Commun* **2024**, 15, 10589, *J. Am. Chem. Soc.* **2023**, 145, 51, 28276–28283, *Sci. Adv.* **2020**, 6, eaaz2060), the modified equation still has the limitation that it cannot accurately describe the complete photodynamic process, and it only provides a reference for the theoretical screening trend. Therefore, we have made comprehensive revisions

to the relevant contents of the original manuscript (Fig. 2, Supplementary Figs. 4-7, Supplementary Figs. 24 and 49, line 90-92, 127-129, 461).

2. The expression of the energy calculation object in equation 3

We are very sorry for the misunderstanding of the energy calculation object in equation 3 due to our inaccurate expression. Thank you for pointing out this problem in time.

In the actual calculation process, the total energy of the complete slab model of the catalyst is always used for calculation, rather than only the surface energy. The expression of this calculation equation strictly follows the reference: *Sci. Adv.* **2020**, 6, eaaz2060, *Angew. Chem. Int. Ed.* **2025**, 64, e202506072. In the original manuscript, "overall slab energy" is expressed as "surface energy", which is a habitual and inappropriate expression to cause the possible ambiguity. In the revised manuscript, we will formally correct the expressions in equation 3 to $\Delta E_{V_S} = E_{\text{slab-V}_S} - E_{\text{slab}} - E_S$, in which $E_{\text{slab-V}_S}$ and E_{slab} represent the energy of the slab after and before V_s , and verify the relevant expressions throughout the manuscript to avoid similar ambiguities (line 454-456).

3. The detail regarding the calculation of the free energy barrier

All the supplementary calculation details mentioned above will be elaborated in detail in the "Density functional theory (DFT) calculations" section of the revised manuscript (line 436-446) and Supplementary Table 1 of the revised supplementary information, and the detailed contents are as follows:

DFT calculations were carried out in the framework of the Vienna ab initio Simulation Package (VASP). The exchange and correlation energies were established by the Perdew-Burke-Ernzerhof (PBE) functional with spin-polarized generalized gradient approximation (GGA) (*Phys. Rev. Lett.* **1996**, 77, 3865). The core electrons were described by the projected augmented wave (PAW) pseudopotentials. For geometry optimization, a cutoff energy of 450

eV was used to expand the wave functions, and the Brillouin zone was sampled with $2 \times 2 \times 1$ k-points. All the structures and energy were allowed to relax below $0.05 \text{ eV } \text{\AA}^{-1}$. The DFT-D3 method of Grimme et al. was employed to involve the van der Waals interaction (*J. Comput. Chem.* **2006**, 27, 1787–1799). The calculations of charge density difference were employed to analyze the movement and distribution of the charge. For the elementary reaction barriers, the transition states (TSs) were determined by the climbing image nudged elastic band (CINEB) method and were confirmed by further frequency calculations showing one and only one imaginary frequency.

Table R1 Calculated free energy for reaction process and corrections for intermediates

Elementary steps	Intermediate	ZPE/eV	TS/eV	G/eV
*+CO ₂ → *CO ₂	*CO ₂	0.31	0.18	0.17
*CO ₂ + H ⁺ + e ⁻ → *COOH	*COOH	0.6	0.22	0.56
*COOH + H ⁺ + e ⁻ → *CO + H ₂ O	*CO	0.19	0.15	-0.14
*CO + H ⁺ + e ⁻ → *CHO	*CHO	0.48	0.15	0.51
*CO + *CHO → *COCHO	*COCHO	0.72	0.19	-0.51
*COCHO + H ⁺ + e ⁻ → *COCHOH	*COCHOH	1.04	0.24	-0.15
*COCHOH + H ⁺ + e ⁻ → *CHOCHOH	*CHOCHOH	1.06	0.2	0.09
*CHOCHOH + H ⁺ + e ⁻ → *CHOCH + H ₂ O	*CHOCH	0.84	0.12	-0.27
*CHOCH + H ⁺ + e ⁻ → *CHOHCH	*CHOHCH	1.17	0.11	-0.19
*CHOHCH + 2H ⁺ + 2e ⁻ → *CH ₂ CH + H ₂ O	*CH ₂ CH	1.26	0.08	-0.89
*CH ₂ CH + H ⁺ + e ⁻ → *CH ₂ CH ₂	*CH ₂ CH ₂	1.38	0.06	-1.24

We sincerely appreciate the reviewer’s time and effort in evaluating our manuscript, as well as the insightful comments and constructive suggestions. We have carefully addressed all concerns raised, and hope the revisions will meet with approval.

Reviewer #2 (Remarks to the Author):

The authors addressed my comments.

Response: We appreciate the reviewer's effort in reviewing our manuscript, positive recommendation and insightful comments very much.

Reviewer #3 (Remarks to the Author):

Response: We appreciate the reviewer's effort in reviewing our revised manuscript and positive recommendation very much.

Reviewer #4 (Remarks to the Author):

Revised ms. which contains new experimental data and clarifies many points is recommended for publication.

Response: We sincerely appreciate the reviewer's time and thoughtful evaluation of our manuscript, which have been invaluable in enhancing the quality of this work.

Reviewer #5 (Remarks to the Author):

The authors have already provided reasonable answers to all the questions that were raised.

Response: We appreciate the reviewer's effort in reviewing our revised manuscript and positive recommendation very much.

Response to Reviewers' Comments

Manuscript number: NCOMMS-25-62587B

Title: Near-unity CO₂-to-ethylene photoconversion over low coordination single-atom catalysts

REVIEWER COMMENTS

Reviewer #1 (Remarks to the Author):

The authors have provided reasonable replies to all my queries and have appropriately revised the manuscript accordingly. Now I do not have any further questions regarding the manuscript.

Response: We sincerely appreciate the reviewer's time and thoughtful evaluation of our manuscript, which have been invaluable in enhancing the quality of this work.

The paper reports on the $\text{Mn}_1\text{-ZnS}_v$ photocatalyst for CO_2 reduction to C_2H_6 . The study employs theoretical guidance in experimental design, and the overall structure is complete, with clear logic and theoretical foundations. However, the following points require further discussion:

1. On page 5, the statement “The low-coordination Mn site in $\text{Mn}_1\text{-ZnS}_v$ strengthens the Mn–CO π backbonding” – what does this mean? Please provide a detailed explanation. What is the relationship between this and the differential charge density analysis?
2. To ensure greater rigor, in addition to calculating the adsorption of $^*\text{CO}$ on the Mn atom in both $\text{Mn}_1\text{-ZnS}$ and $\text{Mn}_1\text{-ZnS}_v$ structures, the authors should also consider the adsorption on Zn atoms surrounding the Mn site. Although $^*\text{CO}$ does not readily adsorb on Zn atoms in pure ZnS, the electronic structure modulation induced by Mn may alter the adsorption behavior on neighboring Zn atoms.
3. On page 6, why can microwave radiation directionally induce defects? What advantages does microwave irradiation offer over other synthesis methods specifically for ZnS materials? How does H_2O_2 etch S? Is it possible that O atoms are introduced into ZnS during this process?
4. In Figure 3e, how can it be confirmed that the bond is Mn–S rather than Mn–O?
5. In Figure 3c, given that the atomic masses of Mn and Zn are very similar and the $\text{Mn}_1\text{-ZnS}_v$ is bulk rather than ultrathin sheet-like, it is generally challenging to distinguish them via ADF-STEM. Can the authors provide a reasonable explanation for this observation?
6. Please include the energy band structures related to other possible CO_2 reduction products to further support the claim that CO_2 is more readily reduced to C_2H_6 .
7. Since water is present in the photocatalytic reaction system, were any products detected in the aqueous solution?
8. It would be beneficial if the authors could calculate the consumption and balance of photogenerated carriers throughout the reaction to further elucidate the stability of the catalyst.